# High expression of interleukin-18 receptor alpha correlates with severe respiratory viral disease and defines T cells with reduced cytotoxic signatures

Aira F. Cabug[1], Jeremy Chase Crawford [2,3], Hayley A. McQuilten [1], Isabelle J. H. Foo [1], Lilith F. Allen[1], Deborah Gebregzabher [1], Robert C. Mettelman[2], Tanya Novak [4], Janet Chou[5], Louise C. Rowntree [1], Ruth R. Hagen [1], Abby J. Thomson[1], Genevieve E. Martin [1,6], Brad Gilbertson [1], Michael NT Souter [1], Fiona James [6], Emma Goodall[6], Simone Rizzetto [7], Tim Flerlage[8], Xiaoxiao Jia[1], Lee-Ann Van de Velde [2], So Young Chang[1], Fabio Luciani[7], Ryan S. Thwaites [9], Jason A. Trubiano[10,11,12,13], Tom C. Kotsimbos[14,15], Allen C. Cheng[16,17], Adrienne G. Randolph [4,18], Paul G. Thomas [2,18], Jianqing Xu [19], Zhongfang Wang[1,20], Thi H. O. Nguyen[1,21], Brendon Y. Chua [1,21], Lukasz Kedzierski [1,21] & Katherine Kedzierska [1,18,21] ✉

Hyperactivated immunity underpins severe outcomes of respiratory viral infections, yet specific immune perturbations are ill-defined. Our recent findings identified OLAH (oleoyl-ACP-hydrolase) as a driver of life-threatening viral diseases. In the same patient cohorts, we now identify the gene encoding IL-18Rα chain (*IL18R1*), as being highly expressed in life-threatening influenza, COVID-19, RSV and multisystem inflammatory syndrome in children (MIS-C) and demonstrate markedly elevated surface protein IL-18Rα expression on CD8 T cells in these infections. Using a mouse model of severe influenza, we further show that high IL-18Rα expression on effector T cells is associated with increased disease severity. We find that IL-18Rα expression on CD8 T cells is inversely associated with cytotoxicity-related genes, including granzyme A, granzyme B, perforin, Eomes, and KLRG-1. Our study demonstrates that IL-18Rα is associated with severe and fatal respiratory disease outcomes and proposes the use of IL-18Rα as a potential biomarker for severe respiratory viral disease.

Viral infections constitute a continuing global threat to human health, yet it is unclear why some individuals develop severe and fatal disease, while others have mild symptoms or remain asymptomatic. Life-threatening influenza and COVID-19 are characterised by hypercytokinemia, over-activation of innate and adaptive immune cell populations, and tissue damage[1–4]. Increased susceptibility to respiratory virus infections is multifactorial and includes impaired or over-active immune responses, minimal pre-existing immunity, and host genetic factors[3,5,6], including single-nucleotide mutations within *IFITM3*, germline mutations of genes associated with TLR3 and

regulatory elements mediating induction of IFN type I and III production[1,2,7–10].

Influenza disease severity is particularly devastating when a new influenza virus capable of infecting humans emerges, as exemplified by avian influenza viruses A(H7N9)[11] and A(H5N1)[12] causing high (>35%) mortality rates in humans. Following the 2013 outbreak of a novel avian A(H7N9) influenza virus in China, we identified immune responses contributing to fatal A(H7N9) disease[2,13,14]. Our longitudinal analyses in patients hospitalized with A(H7N9) demonstrated that individuals with fatal outcomes exhibited hypercytokinemia and perturbed immune responses, in contrast to patients who recovered from severe A(H7N9) disease[2,13]. Importantly, we identified oleoyl-ACP-hydrolase (OLAH), an enzyme involved in endogenous fatty acid production, as an early driver of fatal disease[15]. In this study, using the same patient cohorts, we identified *IL18R1*, the gene encoding the IL-18 receptor (IL-18R) α chain, as a key immune factor associated with fatal A(H7N9) disease and hypercytokinemia. Compared to patients who recovered, patients who later succumbed to A(H7N9) infection had significantly elevated expression of *IL18R1*.

IL-18R comprises of two receptor chains, IL-18Rα and IL-18Rβ, which combine upon ligand-induced signalling[16]. IL-18Rα (CD218α or IL-1R5) serves as the ligand-binding domain and binds IL-18 with low affinity. Upon binding of IL-18 to IL-18Rα, the IL-18Rβ chain is recruited to form a signalling complex, resulting in enhanced binding affinity and initiating downstream signalling through the MyD88-dependent activation of NFκB and members of the MAP kinase family[17]. IL-18Rα is expressed on both immune and non-immune cells, including B cells, macrophages, dendritic cells, neutrophils, T cells and basophils[18]. Expression of *IL18R1* can be high in IFNγ-producing Th1 cells, γδ T cells, NK cells, NK T cells and MAIT cells[19]. Its cognate ligand, IL-18, is a pleiotropic cytokine capable of exerting both pro- and anti-inflammatory effects[20]. The outcome of IL-18 signalling depends on the co-secreted cytokine milieu. Together with IL-12 or IL-15, IL-18 induces pro-inflammatory signals and promotes release of IFNγ[16]. In the absence of IL-12/IL-15, IL-18 drives type II immune responses, stimulating the release of IL-13 and IL-14 from mast cells and basophils[18]. Although the role of IL-18 as both a driver and suppressor of inflammation has been studied, little is known about the role of IL-18R in respiratory viral infections.

In this study, we investigate the underlying factors driving the association between IL18R1 expression and disease severity. We show that surface IL-18Rα expression is increased on CD8 T cells in patients hospitalized with severe influenza A virus, RSV and acute or previous SARS-CoV-2 infection. Furthermore, using a mouse model of severe influenza disease, we demonstrate that high IL-18Rα expression on effector T cells is associated with exacerbated disease outcomes and identify CD8 T cells with enhanced IFNγ production, but reduced cytotoxic effector molecules. Collectively, our study shows how expression of IL-18Rα is associated with severe and fatal respiratory viral disease and proposes its potential utility as a predictive biomarker of severe disease.

## Results

### High expression of *IL18R1* during early avian A(H7N9) influenza, SARS-CoV-2 and RSV infection is strongly associated with severe disease outcome

Our previous whole blood transcriptome analysis in patients hospitalized with severe A(H7N9) infection in China in 2013 (Fig. 1a) identified *OLAH* as the early driver of fatal disease outcomes[15]. Here, in the same A(H7N9) cohort, our analysis of differentially expressed genes identified the *IL18R1* gene, encoding for IL-18Rα receptor (CD218a; IL-18Rα), as one of the three key genes highly expressed in fatal H7N9 patients early after hospital admission (within 6 days of hospital admission), alongside *OLAH* and *FLT3* (Fig. 1b). Importantly, high *IL18R1* levels in patients with fatal A(H7N9) infection were detected early after hospital

admission and persisted until the patients died. In contrast, A(H7N9)-infected patients who recovered had lower *IL18R1* levels during their hospital stay (Fig. 1c; *P* < 0.001).

To determine whether the observed association between *IL18R1* levels and acute A(H7N9) disease severity could also be recapitulated in other respiratory diseases, we assessed *IL18R1* gene expression in a second disease cohort of patients hospitalized with acute SARS-CoV-2 infection, MIS-C (multisystem inflammatory syndrome in children) or paediatric acute respiratory distress syndrome (pARDS) (Fig. 1a). Bulk RNA sequencing analysis of blood from infants, children or young adults (aged 0-21) infected with SARS-CoV-2 showed that *IL18R1* gene expression levels gradually increased with disease severity (Fig. 1d). In comparison to samples from a cohort of healthy individuals that were used as controls (Fig. 1a), *IL18R1* expression was significantly elevated in patients hospitalized with COVID-19 with no or minimal respiratory dysfunction (*P* = 0.019). Relative to milder infections, *IL18R1* expression was elevated in patients with moderate to severe (*P* = 0.029) and life-threatening respiratory dysfunction (*P* = 0.0012) as well as those with MIS-C (*P* < 0.0001), a post-infection severe hyperinflammatory complication linked to prior SARS-CoV-2 infection.

In our third unrelated disease cohort of paediatric acute respiratory cohort[21], single-cell RNA sequencing (scRNAseq) data were obtained from tracheal aspirates collected from children hospitalized with acute respiratory failure requiring endotracheal intubation, stemming from lower respiratory tract infection (LRTI) (Fig. 1a). Analysis of these data similarly demonstrated that *IL18R1* expression was increased in the context of LRTI and pARDS, particularly in comparison to patients who were intubated due to neurologic failure but without diagnosed LRTI or lung disease (Fig. 1e, left). Analysis of the subset of samples collected within 24 hours after intubation demonstrated that *IL18R1* expression was elevated early within the course of the study and particularly high among the patients diagnosed with RSV and the most severe respiratory dysfunction (RSV + P; Fig. 1e, right).

To contextualize *IL18R1* expression levels in healthy individuals and in non-hospitalized patients with mild respiratory infections, we analysed blood microarray data obtained from human challenge models of H1N1 infection, H3N2 infection, human rhinovirus virus (HRV) infection and RSV infection during 38 hours prior to infection and up to 170 hours after infection[22]. These analyses clearly demonstrated unaltered *IL18R1* expression following sham infection and mild infection across any of these mild respiratory virus infections (Fig. 1f). Overall, *IL18R1* expression from human samples obtained up to 7 days after challenge exhibited no differences between sham and mild infection.

In contrast to increased *IL18R1* expression, we found no significant differences in *IL15RA* and *IL12RB1* levels between fatal and recovered A(H7N9) patients across both early and late time points (Supplementary Fig. 1a). Similarly, *IL15RA* levels remained unchanged across disease severities in hospitalized SARS-CoV-2 and MIS-C patients compared to healthy individuals (Supplementary Fig. 1b), in contrast to *IL18R1* expression (Fig. 1d). However, *IL12RB1* and *IL12RB2* transcript levels as well as those encoding for the cytokines, IL15 and IL12 (*IL15* and *IL12A*, respectively) decreased in life-threatening COVID-19 and MIS-C (Supplementary Fig. 1c).

Taken together, our data provide evidence for highly elevated *IL18R1* expression levels, in severe human respiratory viral infections, caused by influenza viruses, RSV and SARS-CoV-2, and life-threatening post-infectious hyperinflammatory MIS-C.

### Increased surface protein expression of IL-18Rα on human CD8 T cells during severe respiratory disease correlates with high *OLAH* levels

Following our findings that high *IL18R1* expression correlated with disease severity at the transcriptomic level, we next investigated whether high surface protein IL-18Rα expression on immune cells was

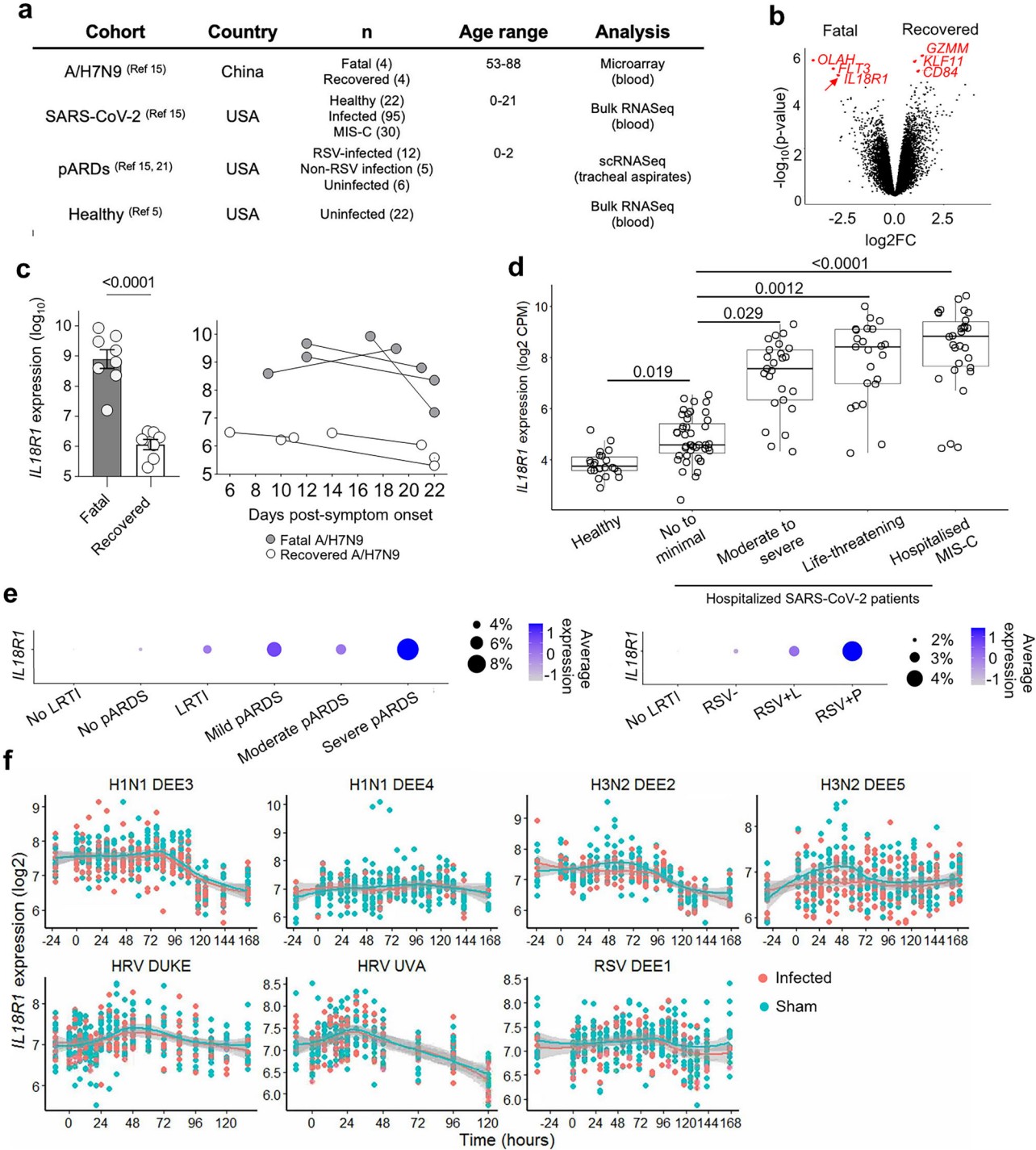

**Fig. 1 | High *IL18R1* expression in patients with fatal H7N9 infection and with life-threatening complications of SARS-CoV-2 and RSV infection. a** Summary of disease cohorts, group sample sizes, age range and generated data used for analysis. **b** Volcano plot showing DEGs between fatal (left) and recovery (right) groups from whole blood microarray of early infection phase samples. **c** *IL18R1* transcript expression levels ($n = 4$ per group at early and late timepoint, mean ± SEM) across early and late time points in A/H7N9 patients. (**d**) Bulk RNA sequencing on whole blood from healthy volunteers and hospitalized patients with acute COVID-19 or MIS-C, aged <21 years ($n = 143$). *IL18R1* expression levels in healthy individuals ($n = 22$) and hospitalized SARS-CoV-2 patients: no/minimal respiratory dysfunction ($n = 43$), moderate/severe ($n = 25$), life-threatening $n = 23$, diagnosed with MIS-C, as described[76] ($n = 30$). *P* values were obtained from a model that controlled for days since symptoms onset, sex, whether a patient was previously healthy, steroid administration prior to sampling, bacterial co-infection, age, race and ethnicity, and

were adjusted for multiple comparisons. **e** *IL18R1* expression in single-cell RNA transcriptomic data acquired from tracheal aspirate samples obtained from RSV-infected children with no/mild ($n = 5$) or moderate/severe pARDS ($n = 7$) (left), and children with non-RSV-related infection but with moderate/severe pARDS ($n = 5$) as well as control children without acute LRTI or lung injury ($n = 6$) (right). Box plot hinges are the first and third quartiles (the 25th and 75th percentiles). The line between hinges corresponds to the media. Whiskers show ± 1.5 * IQR from the hinge (where IQR is the inter-quartile range) **f** *IL18R1* expression across time for human challenge models of mild respiratory infections (H1N1 DEE3 $n = 477$, H3N2 DEE2 $n = 355$, HRV Duke $n = 471$, RSV DEE4 $n = 420$)[22]. Smooth curves fit using a LOESS model. Shaded bands show the 95% confidence interval. Linear mixed models were used to test for effects of time, infection status, and the interaction of time with infection status, with subject included as a random effect. Infection status was not significant for any study.

a hallmark of severe disease outcomes in patients hospitalized with respiratory viral diseases. We analysed surface IL-18Rα expression on CD8 T cells, CD4 T cells and monocytes in the blood from 43 hospitalized RSV, COVID-19, influenza A and influenza B patients during their hospital stays, in comparison to IL-18Rα levels detected in healthy individuals (Supplementary Table 1,2, Supplementary Fig. 2a). Human protein data verified our transcriptomic findings and showed markedly increased surface IL-18Rα expression in patients with acute respiratory viral diseases, as compared to healthy individuals, on CD8 T cells (Fig. 2a), CD4 T cells (Fig. 2b), and to a lesser extent monocytes (Supplementary Fig. 3a). This was irrespective of whether the patients were admitted to the General Ward or Intensive Care Unit (ICU), whether oxygen support was required, or, when analysing influenza A-only patients against healthy individuals (Fig. 2a, b). Increase in IL-18Rα expression in patients was supported by co-expression of activation markers CD38 and HLA-DR on CD8 T cells (Supplementary Figs. 2b, 3b). No correlations between expression of IL-18Rα and IL-18 cytokine levels were found (Supplementary Fig. 4).

To determine whether IL-18Rα expression was further increased in influenza-specific CD8 T cells compared to the bulk CD8 T cell population in hospitalized influenza-A patients, we performed tetramer-associated magnetic enrichment on a subset of HLA-A2[+] patients ($n = 10$) to enrich for influenza-specific CD8 T cells recognizing the immunodominant A2/M1$_{58}$ epitope (Supplementary Fig. 2c). Indeed, influenza-specific CD8 T cells had higher IL-18Rα expression (mean 77.0%) compared to bulk CD8 T cells (59.3%) (Fig. 2c, $P = 0.0068$), whereas there was no difference in IL-18Rα expression for unrelated CMV-specific CD8 T cells (69.9%). Expression of IL-18Rα increased concurrently with additional activation markers such as CD38, HLA-DR or PD-1 in influenza-specific CD8 T cells (Supplementary Figs. 3c, 2d).

To further support our findings, we performed an in-depth analysis of IL-18Rα expression during severe influenza A in a patient case with a prolonged hospital stay (Fig. 3a). The patient was female, aged between 45-49 years old, and had no prior vaccination history or significant comorbidities relating to organ function. The patient was admitted to the Emergency Department (ED) and ICU on day 17 post disease onset. As a result of the recovery, the patient transitioned to the General Ward on day 29 and was subsequently discharged from the hospital on day 35 post-symptom onset. Blood samples were obtained on days 17, 21 and 24 (ICU) and day 31 (General Ward). Analysis of the patient's plasma inflammation showed markedly elevated cytokine levels on hospital/ICU admission on d17 (Fig. 3b), especially evident by massive levels of IL-6, in line with IL-6 concentrations found in patients with fatal avian H7N9 influenza[23] or life-threatening seasonal influenza[3]. Inflammation gradually decreased during the hospital stay and was negligible prior to patient's hospital discharge (Fig. 3b).

By assessing surface protein IL-18Rα expression on CD8 T cells, CD4 T cells and monocytes, in our influenza A patient case, we showed elevated MFIs and frequency of IL-18Rα$^{hi}$ expression on T cells at the time of ICU admission (Fig. 3c, d). As the patient recovered, MFIs and frequency of IL-18Rα expression on T cells were greatly reduced, together with those of monocytes, supporting that IL-18Rα expression correlates with severe respiratory viral disease, as suggested in our RNAseq data in human cohorts (Fig. 1).

In sum, our human data propose that IL-18Rα could serve as a potential biomarker for severe disease.

## Increased IL-18Rα on T cells during influenza virus infection in a mouse model

As the association between IL-18Rα and life-threatening respiratory disease outcomes has not been previously described, we sought to understand the role of IL-18Rα in disease outcomes in a well-characterised C57BL/6 mouse model of influenza A virus (IAV) infection. Mice were infected intranasally (i.n.) either with a $2 \times 10^4$ pfu or $10^3$ pfu dose of A/HKx31 (H3N2) virus to model severe and mild disease,

respectively (Supplementary Fig. 5a)[24]. Mice infected with the severe A/HKx31 dose displayed increased morbidity, as demonstrated by significantly lower body weight at 5, 6 and 7 days post-infection (dpi), when compared to the animals infected with a mild A/HKx31 dose (Area Under the Curve; AUC$_{severe}$ = 1260; AUC$_{mild}$ = 1337, $P < 0.0001$) (Supplementary Fig. 5b, left). Mortality rates were also significantly different ($p = 0.001$), with 50% of mice in the severe group succumbing to infection by 7 dpi (Supplementary Fig. 5b, right). Additionally, severe influenza virus infection was associated with hypercytokinemia, including increased cytokine (IFNβ, TNF, IL-10 and IL-12p70) and chemokine (MCP-1 (CCL2), KC (CXCL1), IP-10 (CXCL10) and RANTES (CCL5)) levels in the severe infection group at 7 dpi (Supplementary Fig. 5c). Thus, our model of severe influenza virus infection accounts for cytokine dysregulation, a common hallmark of highly-pathogenic IAV infections, including A/H7N9, and A/H5N1[2,25] as well as COVID-19[4].

To determine kinetics of IL-18Rα expression on immune cells during IAV infection, we analysed IL-18Rα levels on innate/innate-like and adaptive immune cells in the lung on 1, 3, 6, 10 and 28 dpi (Fig. 4a, Supplementary Fig. 6, Supplementary Fig. 7ab). In naïve mice, IL-18Rα was expressed by a small fraction of conventional CD8 and CD4 T cells (9-14%) but was expressed by a majority of γδ T cells (80%) and to some extent on NK cells (28%) (Supplementary Fig. 7a). Following influenza virus infection, IL-18Rα on CD4 and CD8 T cells was steadily upregulated from the onset of infection, peaking at 6 dpi before gradually returning to baseline levels by 28 dpi (Fig. 4a, Supplementary Fig. 7a, b). Interestingly, comparison of IAV-induced IL-18Rα upregulation on CD4 and CD8 T cells showed that CD8 T cells express significantly higher levels of IL-18Rα at 6 dpi ($P < 0.0001$), despite comparable levels at baseline.

We also observed an increase in IL-18Rα expression on NK T cells at 1 and 3 dpi, although expression levels declined by 6 dpi (Fig. 4a). Conversely, expression levels on γδ T cells and NK cells, although high at baseline, gradually decreased during the acute phase of IAV infection, before returning to steady state levels by ∼day 10 after infection. Minimal IL-18Rα was detected on B cells from naïve or IAV-infected mice across all timepoints following infection (Fig. 4a; Supplementary Fig. 7a). IL-18Rα levels on innate immune cells of the myeloid lineage, including macrophages (both alveolar and interstitial, classical and non-classical monocytes), neutrophils, and dendritic cells (DCs) were consistently low, both before and after IAV infection (Fig. 4a).

## High IL-18Rα levels on T cells correlate with increased influenza disease severity

Given that IL-18Rα expression was upregulated in A/H7N9-infected patients with fatal disease outcomes, we next asked whether IL-18Rα expression levels were associated with disease severity. Indeed, we identified a significant positive correlation between IL-18Rα on CD8 and CD4 T cells and morbidity defined as body weight loss (CD4: $r^2 = 0.8853$, p < 0.0001; CD8: $r^2 = 0.4419$; p < 0.01) (Fig. 4b). Both CD8 and CD4 T cells increased to a significantly higher level early (day 3) during severe IAV infection, when compared to the lower IAV dose (Fig. 4c, Supplementary Fig. 7b). Further analysis of CD4 and CD8 T cell subsets based on CD62L and CD44 activation markers surface staining revealed that the correlation we observed between body weight loss and IL-18Rα was driven by activated CD44$^{hi}$CD62L$^{lo}$ CD8 T cells ($r^2 = 0.7540$, $P < 0.0001$) (Fig. 4d), while a significant correlation was observed for both CD44$^{hi}$CD62L$^{hi}$ and CD44$^{hi}$CD62L$^{lo}$ CD4 T cells (CD44$^{hi}$CD62L$^{hi}$: $r^2 = 0.5359$, $p = 0.0008$; CD44$^{hi}$CD62L$^{lo}$: $r^2 = 0.7367$; $p < 0.0001$) (Supplementary Fig. 7c). Weaker correlations between body weight loss and IL-18Rα were observed for B cells (Supplementary Fig. 7d), NKT cells (Supplementary Fig. 7e), pan dendritic cells and non-classical monocytes (Supplementary Fig. 7f). Accordingly, comparison of the phenotype of IL-18Rα$^+$ and IL-18Rα$^-$ CD8 T cell subsets demonstrated significantly higher levels of the activation markers PD-1 and CD38 on IL-18Rα$^+$ CD8 T cells in both mild and severe groups (Fig. 4e), indicating the highly activated status of IL-18Rα$^+$ T cells. Our

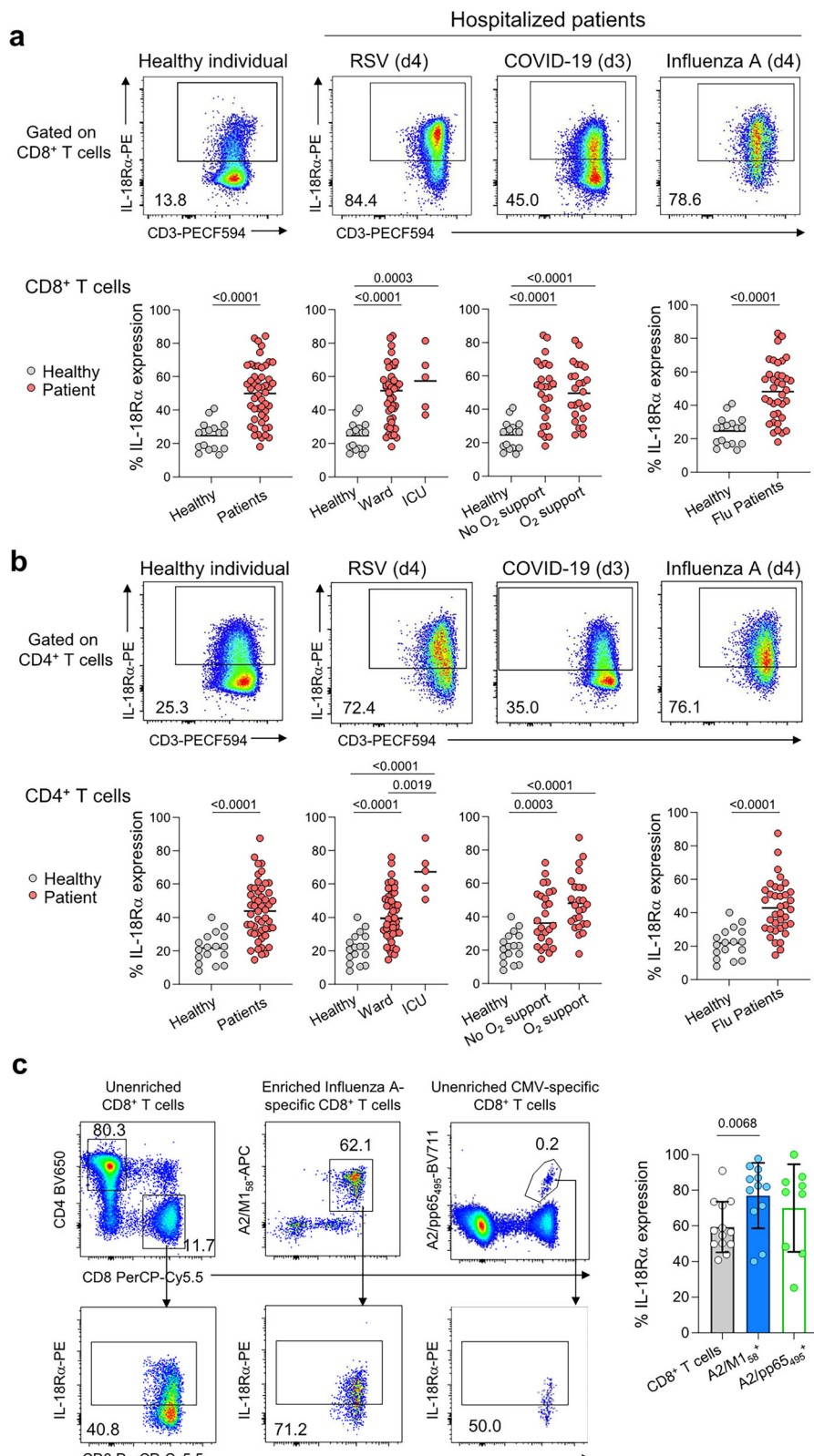

**Fig. 2 | High surface expression of IL-18Rα on human CD8 T cells, CD4 T cells and tetramer⁺ CD8 T cells during severe respiratory disease.** IL-18Rα on human **a** CD8 T cells, **b** CD4 T cells and **c** tetramer⁺ CD8 T cells. **a, b** Representative FACS plots of surface IL-18Rα expression on CD8 T cells and CD4 T cells in a healthy individual and in hospitalized RSV, COVID-19 and influenza A patients are shown; d: days post disease onset. Graphed IL-18Rα expression in healthy individuals (n = 17) and patients hospitalized with influenza A (n = 37), influenza B (n = 1), RSV (n = 4) and COVID-19 (n = 2) at all visit (V) time points, grouped by ICU and ward or oxygen support is shown. IL-18Rα expression in healthy individuals versus influenza A

patients is shown at hospital visit 1 (V1). Bars indicate median. Statistical significance was analysed using a two-tailed Mann-Whitney or Kruskal-Wallis. **c** Representative FACS plots of surface IL-18Rα expression on unenriched CD8 T cells (n = 13), tetramer-enriched influenza-specific A2/M1₅₈⁺CD8 T cells (n = 12) and unenriched CMV-specific A2/pp65₄₉₅⁺CD8 T cells (n = 9) in influenza A patients. Graphed IL-18Rα expression in HLA-A2⁺ patients hospitalized with influenza A at acute timepoints (mean ± SD, two-tailed Wilcoxon matched-pairs signed rank test). Patient demographics are outlined in Supplementary Tables 1 and 2.

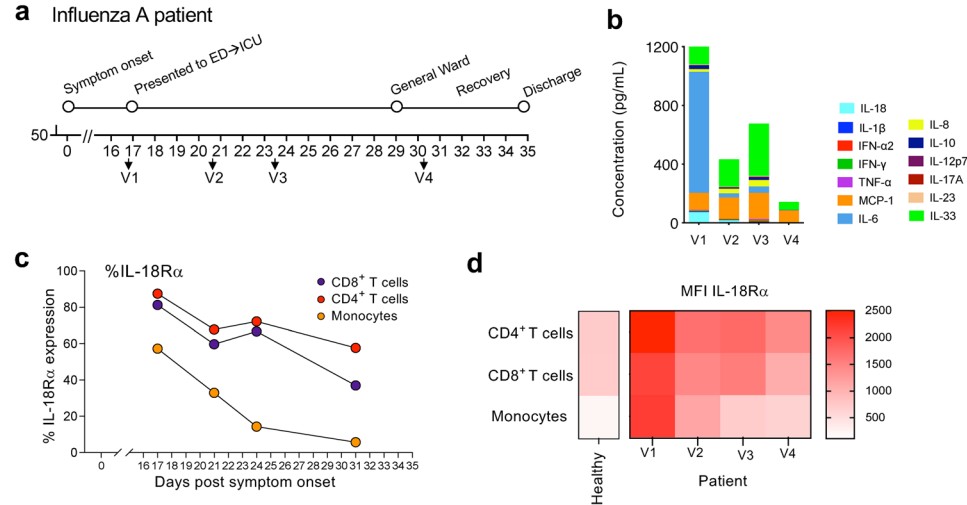

**Fig. 3 | High surface expression of IL-18Rα on T cells in a severe influenza A virus patient case. a** Timeline of influenza A infection, showing hospital admission to the Emergency Department (ED) and intensive care unit (ICU) on day 17 post disease onset (V1), recovery and admission to General Ward on day 29 and discharge on day 31. Blood samples were obtained on days 17 (V1), 21 (V2) and 24 (ICU, V3) and day 31 (General Ward, V4). **b** Proinflammatory cytokines and chemokines in patient sera. surface IL-18Rα expression by flow cytometry, shown as **c** %IL-18Rα expression and **d** mean fluorescence intensity (MFI) on CD8 T cells, CD4 T cells and monocytes in the influenza A patient.

subsequent analysis of the prevalence of CD8 T cells co-expressing IL-18Rα and PD-1 showed that IL-18Rα⁺ PD-1⁺ CD8 T cells were significantly elevated in lungs of the severe group on day 6 (Fig. 4f; $P = 0.0002$), constituting >50% of all the CD8 T cells in nearly all the mice, but minimal IL-18Rα⁺ PD-1⁺ co-expression was displayed on spleen CD8 T cells (Supplementary Fig. 7g). Furthermore, increased frequency of IL-18Rα⁺PD-1⁺ CD8 T cells was associated with more severe disease outcome ($r^2 = 0.8292$, $P < 0.0001$) (Fig. 4g). We also found significant correlations between PD-1, KLRG-1 and both PD-1/CD38 and disease severity on total CD8 T cells (Supplementary Fig. 5d). This is in accordance with persistence of CD8 T cells co-expressing activation markers CD38, PD-1 and HLA-DR in humans linked to a fatal outcome in severe A/H7N9 infection[14]. Here, we used PD-1 as a marker of T cell activation in acute influenza virus infection[26]. It is important to note that it is still unclear whether T cells are truly exhausted during acute influenza virus infection.

Collectively, our data highlight a striking association between activated, antigen-experienced IL-18Rα⁺ CD8 T cells and severe influenza disease, mirroring our previous observations in A(H7N9)-infected patients[14].

### Influenza-specific CD8 T cells display high levels of IL-18Rα during acute and early memory phases

In the C57BL/6 model of primary IAV infection, influenza-specific CD8 T cell responses are directed predominantly towards two immunodominant epitopes, $D^bNP_{366}$ and $D^bPA_{224}$[27,28], which can constitute ~80% of the total influenza-specific CD8 T cell response[24,29]. Therefore, we measured IL-18Rα levels on $D^bNP_{366}$- and $D^bPA_{224}$-specific CD8 T cells on days 10 and 28 following IAV infection (Fig. 4h). High IL-18Rα levels on influenza-specific T cells persisted until at least 28 dpi, while tetramer-negative cells displayed a significant reduction in IL-18Rα levels ($P < 0.0001$) (Fig. 4i, Supplementary Fig. 7h). While the latter may include CD8 T cells directed at minor subdominant IAV specificities, their presence at much lower frequencies[29] suggests that bystander CD8 T cells likely constitute the majority of this tetramer-negative CD8 T cell population. To examine IL-18Rα levels during the secondary recall CD8 T cell response, we challenged mice (either primed with low or high A/HKx31 dose) with a high-dose ($10^5$ pfu i.n.) of A/PR8 (H1N1), a serologically distinct IAV strain. Analysis of influenza-specific CD8 T cells 8 days after the secondary challenge revealed similarly high IL-18Rα levels on $D^bNP_{366}^+$ and $D^bPA_{224}^+$

CD8 T cells, at levels comparable to those observed at day 28 after primary IAV infection (Fig. 4i). Conversely, tetramer-negative CD8 T cells showed upregulation of IL-18Rα, with the average frequency of IL-18Rα⁺ tetramer-negative CD8 T cells nearly doubling from day 28 (1°) until day 8 following the secondary challenge. Interestingly, we observed higher IL-18Rα on $D^bPA_{224}^+$ CD8 T cells in comparison to $D^bNP_{366}^+$ CD8 T cells across all three time points, potentially due to the higher avidity of $D^bPA_{224}^+$ CD8 T cells[28,30]. The specificity of IL-18Rα^hi CD8 T cells that do not stain with the influenza-specific tetramers is unknown, and could potentially reflect bystander CD8 T cells activated in the highly inflammatory milieu. We have analysed the proportion of naïve (CD62L^hiCD44^lo) CD8 T cells within the tetramer-negative population on days 10 and 28 post-primary infection, and day 8 following secondary infection (Fig. 4j). Compared to day 10 and day 8 (2°) post-infection, there is a significantly higher proportion of naïve tetramer-negative CD8 T cells on day 28. These observations are consistent with the fact that a proportion of the tetramer-negative CD8 T cells become activated and transition into CD44^hi effectors during the acute phases of influenza virus infection, resulting in fewer naïve CD8 T cells compared to a memory (day 28) timepoint, during which there is no longer active inflammation or viral replication.

Overall, these findings suggest that while influenza-specific CD8 T cells continue to express high levels of IL-18Rα during the memory phase, bystander T cells only transiently upregulate IL-18Rα during acute IAV infection. Following secondary challenge, both antigen-specific and non-specific T cells upregulate IL-18Rα, likely reflecting both antigen-driven and bystander activation.

Given high levels of IL-18Rα on $D^bNP_{366}$ and $D^bPA_{224}^+$ CD8 T cells at 28 dpi after infection, we subsequently defined IL-18Rα levels across different memory CD8 T cell subsets. Analysis of lung and spleen samples confirmed that both $T_{EM}$ and $T_{CM}$ exhibit significantly higher levels of IL-18Rα at 28 dpi than naïve CD8 T cells ($P < 0.0001$ comparing naïve and $T_{EM}/T_{CM}$ in both organs) (Fig. 4k,l). These findings demonstrate that memory CD4 and CD8 T cells generated upon IAV infection, but not naïve CD4 or CD8 T cells, have high levels of IL-18Rα.

### Transcriptomic analysis reveals key differences between IL-18Rα high and IL-18Rα low antigen-specific CD8 T cells

Having identified a key correlation between IL-18Rα on activated CD44^hiCD62L^lo CD8 T cells in the lung (Fig. 4d) and morbidity at day 6

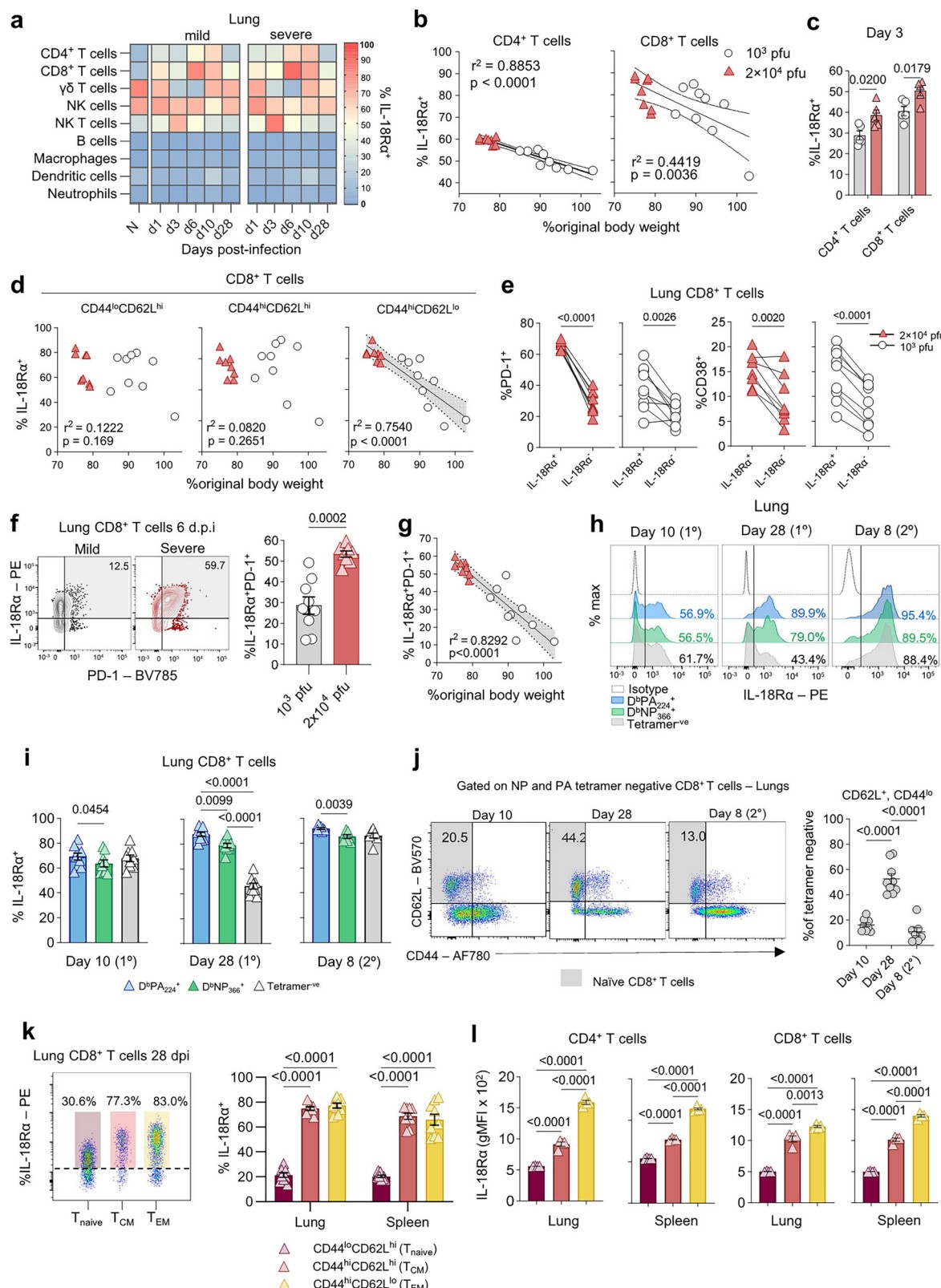

after infection in mice, and in patients hospitalized with respiratory viral diseases (Fig. 2), we further delved into the mechanisms underlying high levels of IL-18Rα on effector CD8 T cells following high dose influenza infection. To define differences between antigen-specific CD8 T cells with high IL-18Rα and low IL-18Rα levels during high dose IAV infection, we performed RNA sequencing using a well-established OVA/OT-I system[31] and infection with a recombinant X31 influenza

virus strain expressing the OVA antigen (HKx31-OVA)[32]. Following the transfer of $10^5$ transgenic CD45.1[+] OT-I cells into congenically-different CD45.2 mice, the animals were infected with a high dose ($10^5$ pfu) of HKx31-OVA virus (Fig. 5a). Six days after infection, lungs were harvested and activated CD44[hi]CD62L[lo] OT-I cells were sorted based on their IL-18Rα levels into IL-18Rα[hi] and IL-18Rα[lo] subsets (Supplementary Fig. 8c). The latter comprised of CD44[hi]CD62L[lo] OT-I cells with

**Fig. 4 | High IL-18Rα expression on αβ T cells correlates with increased influenza disease severity. a** Mean frequency of IL-18Rα+ cells on innate and adaptive immune cells in the lung of naïve mice (N, $n = 5$) and on 1, 3, 6, 10, and 28 dpi in mild ($n = 5$) and severe groups ($n = 4$–5). **b** Correlation between body weight loss and IL-18Rα expression on CD4 and CD8 T cells in the lung, 6 dpi (grey bands show 95% CI). $r^2$ = Pearson's correlation coefficient. Data pooled from 2 independent experiments. **c** IL-18Rα+ T cell frequencies in lungs at 3 dpi ($n = 5$ per group, mean ± SEM, two-tailed unpaired t-test). **d** Correlation between body weight loss and IL-18Rα on CD8 T cells based on CD44 and CD62L expression (Pearson's correlation, 95% CI). **e** PD-1 and CD38 on IL-18Rα+ and IL-18Rα- CD8 T cells in mild ($n = 9$) and severe ($n = 8$) groups (two-tailed paired t-test) (left). Frequencies of IL-18Rα+PD-1+ CD8 T cells in the lung of mild ($n = 9$) and severe ($n = 8$) groups (mean ± SEM, two-tailed Mann-Whitney). **f** Co-expression of IL-18Rα and PD-1 in lungs of mild ($n = 8$) and severe group ($n = 9$) on day (mean ± SEM, two-tailed Mann-Whitney). **g** Correlation between body weight loss and frequencies of IL-18Rα+PD-1+ cells (Pearson's correlation, 95% CI). **h** Representative histograms showing IL-18Rα on influenza-specific and non-specific CD8 T cells during acute (10 dpi) and memory (28 dpi) phases of primary IAV infection ($2 \times 10^4$ pfu A/HKx31), and on 8 dpi following secondary challenge ($10^5$ pfu A/PR8). **i** IL-18Rα on $D^bPA_{224}$ and $D^bNP_{366}$ CD8 T cells and non-specific (tetramer-) CD8 T cells (RM one-way ANOVA, Geisser-Greenhouse and Tukey's correction, $n = 7$–9, mean ± SEM). **j** Frequency of naïve cells within $D^bNP_{366}$ and $D^bPA_{224}$-tetramer negative CD8 T cells ($n = 7$–9, mean ± SEM, Holm-Sidak's one-way ANOVA). IL-18Rα on different memory CD8 T cell subsets in the lung, 28 dpi. IL-18Rα+ **k** frequencies and **l** geometric mean fluorescence intensity (gMFI) in naïve ($T_{naive}$, CD44-, CD62L+), central memory ($T_{CM}$, CD44+CD62L+, light red) and effector memory ($T_{EM}$, CD44+CD62L-, yellow) CD4 and CD8 T cell subsets in lung and spleen, 28 dpi ($n = 4$ per group, RM two-way ANOVA, Geisser-Greenhouse and Tukey's correction, mean ± SEM). Data shown on 6, 10, and 28 dpi are from two independent experiments.

low-to-intermediate levels of IL-18Rα (Fig. 5b). It is important to note that this OT-I experiment was performed to define transcriptomic differences between IL-18Rα^hi and IL-18Rα^lo CD8 T cells subsets following high dose influenza virus infection (the highest approved dose by our ethics) rather than during severe disease, as transferring large numbers of transgenic OT-I cells into mice prevents significant body weight loss following viral infection.

In total, RNAseq identified 237 protein-coding differentially expressed genes (DEGs) distinguishing IL-18Rα^hi and IL-18Rα^lo OT-I populations; 134 were overexpressed in the IL-18Rα^hi subset and 103 in the IL-18Rα^lo set (Fig. 5cd). The DEGs enriched in the IL-18Rα^hi OT-I population included transcription factors involved in CD8 T cell differentiation and development (*Tox, Myc, Junb, Id3, Satb1, Runx2*), T cell survival (*Bcl2a1b, Bcla1d, Tcf7*) and genes linked to the NFκB pathway *(Nfkb2, Nfkbid, Nfkbia, Nfkbiz*) (Fig. 5e). In addition, IL-18Rα^hi OT-I cells expressed higher levels of cytokines and chemokines, including *Ccl3, Ccl1, Cxcl10, Tnfsf11* (LIGHT), *Tnfsf14* (RANKL), the DC chemoattractant *Xcl1*, and notably, *Ifng*, indicating a strong inflammatory potential of IL-18Rα^hi CD8 T cells. Cytokine and chemokine receptors were also differentially expressed between the two groups; the chemokine receptors *Il2ra* and *Il7ra*, associated with promoting the survival of memory CD8 T cells, were higher in IL-18Rα^hi OT-I population. Conversely, *Ccr2, Cx3cr1* and *Il10ra* were higher in IL-18Rα^lo OT-I population.

Although IL-18Rα^hi OT-I cells were generally enriched for genes encoding cytokines, IL-18Rα^lo OT-I cells had higher expression levels of cytotoxicity-related transcripts, including granzymes (*GzmA, GzmB, GzmK*), *Fasl* (Fas ligand) and *Prf1* (perforin) (Fig. 5e). Furthermore, IL-18Rα^hi OT-I cells expressed high levels of genes encoding cytokine/chemokine production and early activation markers such as *Cd69, Tnfrsf9* (4-1BB), *Tnfrsf9* (OX4O), together with *Nrp1, Pdcd1*.

As Notch regulates CD8 T cell effector functions, including expression of Eomes, perforin, GzmB and IFNγ[33], we also analysed genes involved in the Notch pathway (Fig. 5e). In accordance with increased *GzmA, GzmB* and *GzmK* genes in IL-18Rα^lo OT-I cells, we found increased expression of two genes that regulate Notch signalling; *Dtx1*[34] and *St3gal6*[35] in IL-18Rα^lo OT-I cells, although we did not observe increased Notch receptor transcript levels.

Activated T cells switch energy production from oxidative phosphorylation to glycolysis to support the performance of their effector functions, such as IFN-γ production[36]. As IL-18, together with IL-12, can increase the glycolysis[37], we also analysed genes involved in the glycolytic pathway. Genes within this glycolytic pathway, including *Fut8, Aurka, Nt5e, Mif, Pgam1* and *Tpi1*, were increased in IL-18Rα^hi OT-I cells, while *Clrx* and *Cenpa* were expressed in IL-18Rα^lo cells (Fig. 5e).

Overall, identification of DEGs crucial to CD8 T cell differentiation, activation and cytokine/chemokine signalling suggests that high IL-18Rα expression plays an important role in the inflammatory response during severe influenza disease.

## Transcriptome of IL-18Rα^hi CD8 T cells bears a pro-inflammatory signature

To gain a global perspective of how molecular profiles of IL-18Rα^hi and IL-18Rα^lo CD8 T cells relate to their function, we defined key biological processes associated with IL-18Rα^hi and IL-18Rα^lo CD8 T cell populations. The gene network analysis highlighted functional connections between the enriched genes with individual clusters, indicating the enrichment of select pathways and biological processes in IL-18Rα^hi cells related to activation and cytokine/chemokine production (Fig. 5f). Comparison of our gene sets to the hallmark gene sets from the molecular signatures database (MsigDB) revealed that the transcriptome of IL-18Rα^hi OT-I cells had significant enrichment for genes involved in TNF signalling via NFκB, IL-2/STAT5 signalling, inflammatory response, IFNγ response and IL-6/JAK/STAT3 signalling (Fig. 5g), all of which are associated with exacerbated inflammation and the highly activated effector state. In agreement with our transcriptomics data, we found markedly increased capacity of influenza-infected lung IL-18Rα^hi CD8 T cells to produce IFNγ following stimulation with IL-12 and IL-18 overnight, while there was a clear lack of IFNγ production by lung IL-18Rα^lo CD8 T cells following influenza virus infection (Fig. 5h).

Apart from the IL-6/JAK/STAT3 and TNF pathways, several of the processes were also enriched in the IL-18Rα^lo OT-I population, albeit to a much lesser extent. Notch signalling, enriched in IL-18Rα^lo, supports the differentiation of naïve T cells into differentiated effector cells expressing cytotoxic molecules or short-lived effector cells (SLECs) defined by *Klrg1* expression (Fig. 5e).

## Influenza-specific IL-18Rα^hi CD8 T cells express reduced cytotoxic signature with TCF1+KLRG1^lo phenotype

As the genes encoding for the transcription factors Eomes and TCF1 (*Tcf7*), were differentially expressed in IL-18Rα^lo and IL-18Rα^hi OT-I cells in our RNAseq analysis (Fig. 5), we next sought to validate Eomes and TCF1 protein levels in influenza-specific $D^bNP_{366}^+$ and $D^bPA_{224}^+$ CD8 T cells following severe IAV infection (Fig. 6a, Supplementary Fig. 8c). Consistent with our findings from transcriptomic analyses, levels of Eomes were significantly higher in both lung and spleen in the IL-18Rα^lo CD8 T cells of both $D^bNP_{366}$ and $D^bPA_{224}$ specificities, while TCF1 was significantly higher in the IL-18Rα^hi $D^bNP_{366}^+$ and $D^bPA_{224}^+$ CD8 T cells (Fig. 6b), further implicating distinct transcriptional control of IL-18Rα^lo and IL-18Rα^hi CD8 T cell effectors.

At the transcriptomic level, IL-18Rα^lo and IL-18Rα^hi OT-I cells also displayed differential activation genes pointing to distinct effector functions. Thus, we investigated the levels of activation markers associated with these differentially expressed genes at the protein level (Fig. 6c–e). Immunophenotyping of IL-18Rα^hi and IL-18Rα^lo tetramer-specific CD8 T cells confirmed that KLRG1, a hallmark of terminally-differentiated effectors[38], was significantly higher in the IL-18Rα^lo CD8 T cell population (Fig. 6c), whereas NRP1, recently

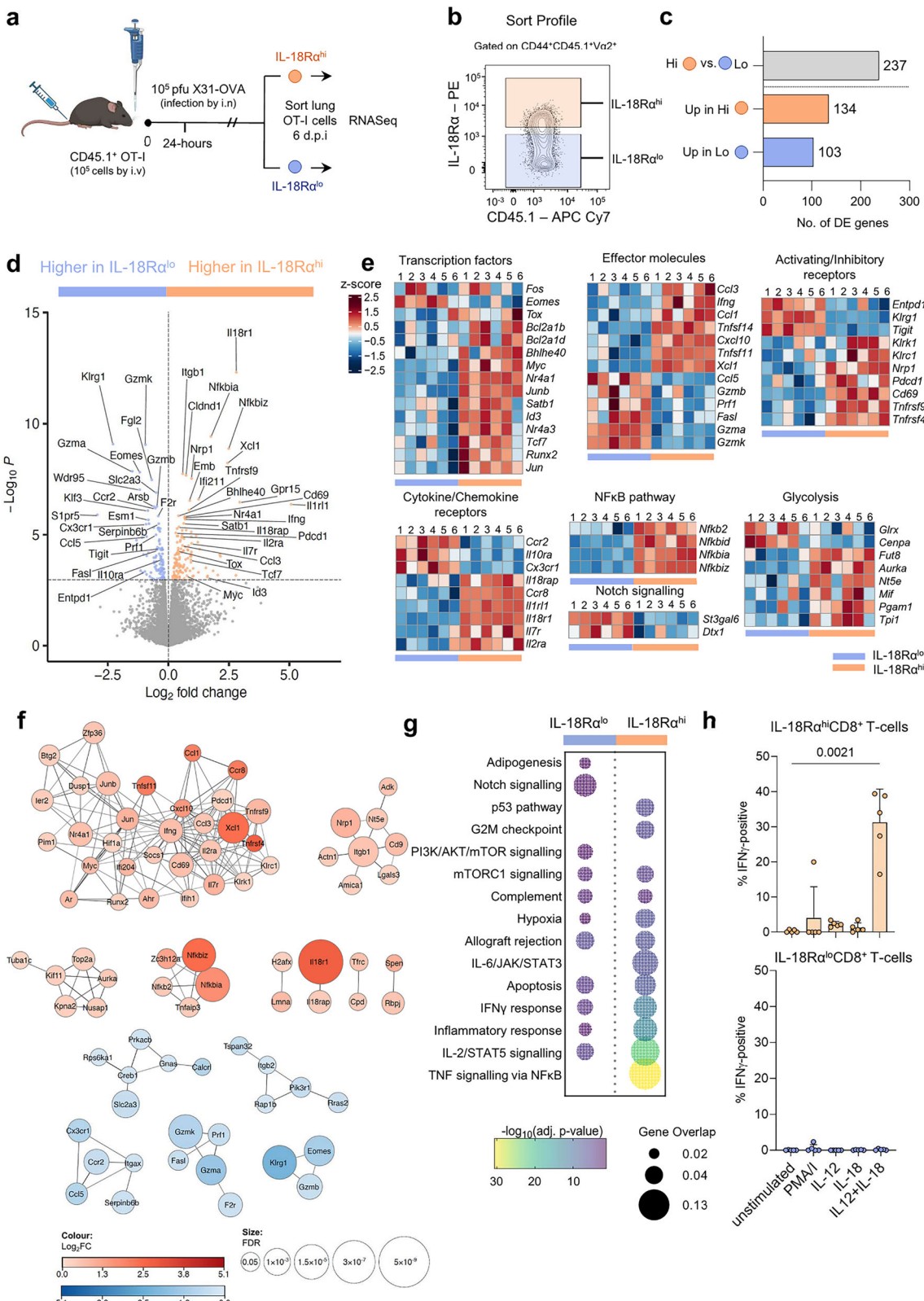

identified as a novel immune checkpoint[39], was significantly increased in the IL-18Rα[hi] CD8 T cell subset on D[b]NP$_{366}$[+] and D[b]PA$_{224}$[+]CD8 T cells (Fig. 6d). The chemokine receptor CX3CR1, which identifies distinct memory CD8 T cell subsets and their precursors[40], was expressed at similar frequencies across IL-18Rα[lo] and IL-18Rα[hi] subsets, although CX3CR1 MFI was increased in IL-18Rα[lo] CD8 T cells (Fig. 6e), suggesting that although there is a similar proportion of CX3CR1[+] tetramer-

specific CD8 T cells in both populations, CX3CR1 is more abundant in the IL-18Rα[lo] subset.

Following our findings of strong cytotoxic gene signatures in IL-18Rα[lo] OT-I cells, we further investigated the capacity of IL-18Rα[hi] and IL-18Rα[lo] CD8 T cells amongst D[b]NP$_{366}$[+] and D[b]PA$_{224}$[+] CD8 T cell subsets to produce cytotoxic effector molecules; granzymes and perforin (Supplementary Fig. 9). Indeed, there was a significantly higher

**Fig. 5 | CD8 T cells expressing high and low levels of IL-18Rα possess distinct transcriptomic profiles. a** $1 \times 10^5$ CD45.1$^+$ OT-I cells adoptively transferred into C57BL/6 mice. Mice were infected with $10^5$ pfu of A/HK x 31-OVA i.n. and lungs were collected on 6 dpi and sorted for cells expressing high (IL-18Rα$^{hi}$) and low-to-intermediate levels of IL-18Rα (IL-18Rα$^{lo}$). Created in BioRender. Cabug, A. (2025) https://BioRender.com/q5po0oj. **b** Antigen-experienced OT-I cells were sorted as CD44$^+$CD45.1$^+$TCRVα2$^+$, then sorted into IL-18Rα$^{hi}$ and IL-18Rα$^{lo}$ subpopulations. **c** Number of differentially expressed genes (DEG) between IL-18Rα$^{hi}$ and IL-18Rα$^{lo}$ OT-I cells. **d** Volcano plot showing the DEGs between IL-18Rα$^{hi}$ and IL-18Rα$^{lo}$ cells. Horizontal line indicates -log$_{10}$(P = 0.05). **e** Heatmaps showing expression levels of select genes grouped according to functional clusters. Each square represents an individual mouse where the blue and orange bars represent paired IL-18Rα$^{lo}$ and IL-18Rα$^{hi}$ cells, respectively ($n = 6$). Z-score values are shown. **f** DEGs in each subpopulation were analysed for protein-protein interaction based on the calculation of a STRING interaction score. Bubble size is FDR and colour intensity represents the magnitude of the difference in gene expression between IL-18Rα$^{hi}$ and IL-18Rα$^{lo}$ subsets (STRING score>0.7). Red networks represent genes more highly expressed in IL-18Rα$^{hi}$ and blue gene networks are those more highly expressed in IL-18Rα$^{lo}$. **g** DEGs were compared to the MSigDB hallmark gene sets. Bubble size corresponds to gene overlap and represents the number of DEGs enriched in gene set. Bubble colours show the -log$_{10}$(adjusted P-value). **h** IFNγ intracellular cytokine staining of IL-18Rα$^{lo}$ and IL-18Rα$^{hi}$ CD8 T cells from lungs on day 6 following influenza virus infection following IL-12, IL-18 or IL-12 + IL-18 stimulation. Unstimulated negative control and PMA/I stimulated positive control were included. IFNγ production by IL-18Rα$^{hi}$ CD8 T cells (top); IFNγ production by IL-18Rα$^{lo}$ CD8 T cells (bottom), ($n = 5$, mean ± SD, one-way ANOVA, Kruskal-Wallis, Dunn's multiple comparisons).

frequency of tetramer-specific IL-18Rα$^{lo}$ CD8 T cells producing granzyme B and perforin, as compared to the IL-18Rα$^{hi}$ CD8$^+$ cells at the site of infection (Fig. 6f–i). Both D$^b$NP$_{366}$ $^+$ and D$^b$PA$_{224}$$^+$ CD8 IL-18Rα$^{lo}$ T cells in lungs also had higher abundance of granzymes A and B (Fig. 6j), confirming lung OT-I RNAseq data (Fig. 4e). In addition, the co-expression profiles of tetramer-specific IL-18Rα$^{lo}$ and IL-18Rα$^{hi}$ CD8 T cells were significantly different in both the lungs and spleen, with a higher proportion of IL-18Rα$^{lo}$ T cells expressing granzymes A, B and/ or perforin (Fig. 6k).

Taken together, these data highlight key differences in the transcriptional, phenotypic, and cytotoxic capabilities of IL-18Rα$^{hi}$ and IL-18Rα$^{lo}$ cells (Fig. 6l). IL-18Rα$^{lo}$ antigen-specific CD8 T cells display higher protein levels of cytotoxic molecules, and the effector markers KLRG1, Eomes and CX3CR1.

### IL-18Rα$^{hi}$ CD8 T cells display increased capacity to produce IFNγ and TNF

To define the polyfunctionality of IL-18Rα$^{hi}$ and IL-18Rα$^{lo}$ subsets of influenza-specific D$^b$NP$_{366}$$^+$ and D$^b$PA$_{224}$$^+$ CD8 T cells, we first assessed the frequencies of IFNγ-producing CD8 T cells following NP$_{366}$ or PA$_{224}$ peptide stimulation in an intracellular cytokine secretion assay[28]. Significantly higher frequencies of IL-18Rα$^{hi}$ CD8 T cells producing IFNγ were detected in both lungs and spleen (lung D$^b$NP$_{366}$$^+$CD8$^+$ $P = 0.00581$; lung D$^b$PA$_{224}$$^+$CD8$^+$ $P = 0.000836$; spleen D$^b$NP$_{366}$$^+$CD8$^+$ $P = 0.000021$; spleen D$^b$PA$_{224}$$^+$CD8$^+$ $P = 0.000001$), compared to IL-18Rα$^{lo}$ CD8 T cells (Fig. 7a; Supplementary Fig. 8d). Furthermore, our analysis of simultaneous production of multiple antiviral cytokines, a hallmark of T cell polyfunctionality and high avidity[41], revealed that a higher proportion of IL-18Rα$^{hi}$ CD8 T cells had the capacity to secrete multiple cytokines, irrespective of their antigen specificity (Fig. 7b-d). There were significantly higher frequencies of IL-18Rα$^{hi}$ CD8 T cells which could produce both IFNγ and TNF (lung D$^b$NP$_{366}$$^+$CD8$^+$ $P = 0.000973$; lung D$^b$PA$_{224}$$^+$CD8$^+$ $p = 0.000008$; spleen D$^b$NP$_{366}$$^+$CD8$^+$ $p = 0.000593$; spleen D$^b$PA$_{224}$$^+$CD8$^+$ $P = 0.000488$) (Fig. 7b) as well as IFNγ$^+$TNF$^+$IL-2$^+$ (lung D$^b$NP$_{366}$$^+$CD8$^+$ $P = 0.003255$; lung D$^b$PA$_{224}$$^+$CD8$^+$ $P = 0.006524$; spleen D$^b$NP$_{366}$$^+$CD8$^+$ $p = 0.000345$; spleen D$^b$PA$_{224}$$^+$CD8$^+$ $P = 0.001407$) (Fig. 7c). Thus, in both the lung and spleen, antigen-specific IL-18Rα$^{hi}$ CD8 T cells displayed increased cytokine capacity comparing to IL-18Rα$^{lo}$ CD8 T cells in terms of their ability to secrete and co-secrete IFNγ, TNF and IL-2 (Fig. 7d). However, it is important to note that IL-18Rα$^{hi}$ CD8 T cells are enriched in antigen-specific CD8 T cells, and this might contribute to such increased cytokine production capacity.

### IL-18Rα$^{hi}$ CD8 T cells predominate at late divisions following in vivo IAV infection

RNAseq data showed that the IL-18Rα$^{hi}$ OT-I cells had enriched transcripts related to T cell activation and expressed higher levels of co-stimulatory molecules related to interactions with antigen-presenting cells (APCs), while the ICS data showed that IL-18Rα$^{hi}$D$^b$NP$_{366}$$^+$ and D$^b$PA$_{224}$$^+$ CD8 T cells produced more cytokines compared to the IL-18Rα$^{lo}$ CD8 T cells, indicating their highly activated state. We further asked whether IL-18Rα levels on antigen-specific CD8 T cells relate to their proliferation capacity following IAV infection in vivo. Using the OT-I transgenic system, we assessed the frequency of IL-18Rα$^+$ and IL-18Rα$^-$ OT-I cells at different T cell divisions following IAV-OVA infection in vivo. Naïve CD45.1$^+$ OT-I cells ($10^6$) labelled with VPD450 were adoptively transferred into CD45.2$^+$ C57BL/6 mice, then 24 hrs later mice were infected with $10^5$ pfu A/HKx31-OVA and levels of IL-18Rα were determined in the draining mediastinal lymph node (mLN) at days 2.5, 3.5 and 4.5 after infection (Supplementary Fig. 10a). Analysis of IL-18Rα expression on OT-I cells at 3.5 dpi in mLN showed upregulation of IL-18Rα with each successive round of division (Supplementary Fig. 10b). While frequencies of IL-18Rα$^+$ OT-I cells increased each day, the IL-18Rα$^-$ OT-I population was significantly higher at every time point analysed, although at 4.5 dpi IL-18Rα$^+$ OT-I cells were as prominent as IL-18Rα$^-$ OT-I cells in division 7+ (Supplementary Fig. 10c). Importantly, the frequency of IL-18Rα$^+$ OT-I cells was significantly higher in later divisions than IL-18Rα$^-$ OT-I cells at all timepoints analysed. Starting at 2.5 dpi, we detected higher frequencies of IL-18Rα$^+$ in divisions 4-7 and by 4.5 dpi almost all IL-18Rα$^+$ cells reached division 7+, while the IL-18Rα$^-$ OT-I population was still actively dividing (Supplementary Fig. 10d). Therefore, it is apparent that the higher activation profile of IL-18Rα$^+$ CD8 T cells relates to the observation that IL-18Rα$^+$ CD8 T cells are generally found at later divisions, which could result from the higher proliferative capacity of IL-18Rα$^+$ CD8 T cells and/or CD8 T cells acquiring the IL-18Rα phenotype at later divisions.

### Human A/H7N9 and COVID-19 transcriptomics reveal inverse correlations between IL-18Rα and cytotoxic T cell signatures associated with severe disease outcomes

To further validate our OT-I gene expression data (Fig. 5) in human cohorts of acute respiratory disease, we analysed our transcriptomics data (Fig. 1[15]) for key genes associated with high expression of IL-18Rα. In both our A/H7N9 (Fig. 8a) and COVID-19 (Fig. 8b) datasets, *IL18R1* expression inversely correlated with expression of *GZMA, GZMB, PRF1* (Perforin 1)*, EOMES, KLRG1* and *CX3CR1*. As *IL18R1* expression in humans (Fig. 1c-e, Fig. 2, Fig. 3) and mice (Fig. 4b) is associated with disease severity, it was not surprising that *GZMA, GZMB, PRF1, EOMES, KLRG1* and *CX3CR1* gene expression decreased with disease severity (Fig. 8a, b). Our analysis of *GZMA*, *GZMB*, *PRF1*, *EOMES*, *KLRG1*, *CXCR1* and *IL18R1* expression levels stratified according to these age ranges showed no clear correlation between age and expression of any of these genes (Supplementary Fig 11).

Our human analyses of key differentially expressed genes from our OT-I experiment (Fig. 4) also demonstrated that high levels of *IL1R1L1* (Fig. 8c) were associated with disease severity, while higher levels of *TIGIT, IL10RA* and *XCL1* were related to the healthy status and milder respiratory viral disease (Fig. 8c).

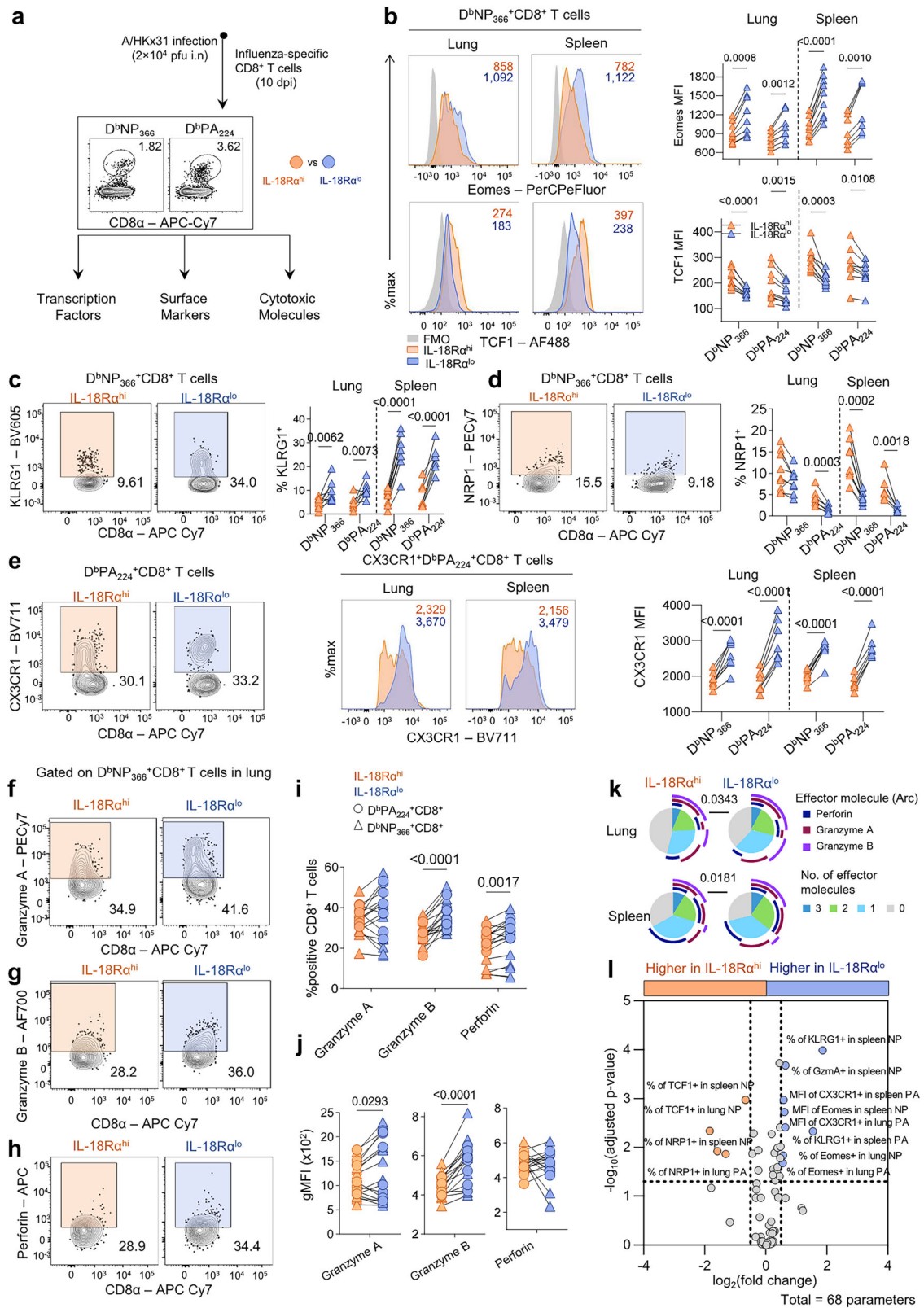

## Discussion

Although respiratory viral infections can cause severe disease, the mechanisms underlying specific immune perturbations are not well understood. Our previous study identified a gene coding for oleoyl-ACP-hydrolase (OLAH) as a key early driver of severe and fatal respiratory viruses[15]. In this study, using the same patient cohorts, we found that the gene encoding for the alpha chain of the IL-18 receptor, *IL18R1*, was also

highly expressed in patients who succumbed to severe A(H7N9) infection, as well as in patients with life-threatening COVID-19, RSV and MIS-C. Our *IL18R1* transcriptomics data were further verified at the protein level and demonstrated markedly elevated surface IL-18Rα expression in patients hospitalized with influenza A, influenza B, RSV and COVID-19 on CD8 T cells. At present, there is a lack of knowledge on the role of IL-18Rα in severe respiratory viral disease, particularly its involvement in

**Fig. 6 | IL-18Rα^lo influenza-specific CD8 T cells are KLRG1^hi exhibit increased production of cytotoxic effectors. a** C57BL/6 mice were infected with $2 \times 10^4$ pfu of A/HKx31, and IL-18Rα^hi (orange) and IL-18Rα^lo (blue) influenza tetramer-specific CD8 T cells were analysed on 10 dpi. **b** Representative histograms (left) showing the expression of Eomes and TCF1 in IL-18Rα^hi and IL-18Rα^lo D^bNP_366-specific CD8 T cells. gMFI values shown on the right. Representative FACS plots and graphs comparing the frequency of IL-18Rα^hi and IL-18Rα^lo tetramer-specific cells expressing (**c**) KLRG1 and (**d**) NRP1. **e** CX3CR1 expression was compared between tetramer-specific IL-18Rα^hi and IL-18Rα^lo cells. Representative populations showing expression of **f** granzyme A, **g** granzyme B and **h** perforin compared between IL-18Rα^hi and IL-18Rα^lo groups. **i** Frequencies of IL-18Rα^hi and IL-18Rα^lo tetramer-specific CD8 T cells expressing cytotoxic molecules in the lungs. Data show pooled D^bPA_224 and D^bNP_366 specificities. **j** gMFI of cytotoxic mediators expressed by IL-18Rα^hi and IL-18Rα^lo tetramer-specific CD8 T cells in the lungs. Data show pooled D^bPA_224 and D^bNP_366 specificities. **k** Polyfunctional profiles (granzyme A, granzyme B, perforin) of pooled tetramer-specific IL-18Ra^lo and IL-18Ra^hi CD8 T cells in the lungs and spleen. Data were analysed via permutation test. **l** Volcano plot summarising differences in expression of all transcription factors, surface markers, and cytotoxic effectors (total $n = 68$ parameters) analysed. Data were analysed via multiple paired t-test with Holm-Sidak's correction for multiple comparisons. The horizontal dotted line is equivalent to $p = 0.05$ and the vertical lines denote a $\log_2$(fold change) of $\pm 0.5$. For (**a–l**), data were analysed using a two-tailed paired t-test where $n = 7$–9 mice per group. Data shown are from two independent repeats.

cell-mediated immune responses. Here, we defined the role of IL-18Rα in acute respiratory viral infections and its potential as a biomarker of severe viral disease.

The IL-18 signalling pathway has previously been implicated as a correlate of severe disease. Indeed, recent studies investigating multisystem inflammatory syndrome in children (MIS-C), a life-threatening post-infectious sequelae of COVID-19 disease in paediatric cohorts, have shown that IL-18 concentrations were significantly increased in the plasma of patients with MIS-C as well as Hemophagocytic lymphohistiocytosis (HLH) relative to healthy controls[42]. Likewise, using a high-dimensional mass cytometry approach to phenotype immune cells in MIS-C, Zhang et al. further demonstrate elevated expression of IL-18R on CD16^+ NK cells, monocytes, as well as TCR Vβ21.3^+CD4^+ and CD8 T cells in MIS-C patients[43].

Here, we further expand on this association between disease severity and the IL-18 signalling pathway and show that high IL-18Rα expression is a feature of multiple severe respiratory infections, including highly pathogenic avian influenza H7N9 and respiratory syncytial virus (RSV), in addition to severe COVID-19. Consistent with previous findings by other groups, we further elaborate that IL-18Rα expression on CD8 and CD4 T cells strongly correlates with disease severity and shows that these can be recapitulated in a mouse model of mild and severe influenza. Collectively, our findings further support the use of IL-18Rα as a potential biomarker for severe respiratory infections.

In our study, we utilised a mouse model of influenza virus infection to investigate the role of IL-18Rα in mild and severe influenza. We found that conventional αβ T cells upregulated IL-18Rα following infection, while NK cells and unconventional T cells downregulated IL-18Rα levels. Furthermore, IL-18Rα levels were significantly elevated in severe influenza disease, and this increase of IL-18Rα on activated T cells at the site of infection strongly correlated with increased disease morbidity. Phenotypic, functional and transcriptomic analyses of influenza-specific OT-I cells with high (IL-18Rα^hi) and low (IL-18Rα^lo) levels of IL-18Rα revealed differential transcriptomic features of IL-18Rα^hi and IL-18Rα^lo T cells. IL-18Rα^hi T cells were pro-inflammatory and polyfunctional, with high expression of *Ifng, Il2ra, Il7ra, Ccl3, Ccl1, Cxcl10, Tnfrsf9, Tnfsf11, Tnfsf14*, genes linked to the NFκB pathway (*Nfkb2, Nfkbid, Nfkbia, Nfkbiz*) and *Tcf7*. Conversely, IL-18Rα^lo T cells had high expression of cytotoxicity-related transcripts, *GzmA, GzmB, GzmK, Fasl* and *Prf1*, with high *Klrg1* and *Eomes* expression. Our study thus defined how IL-18Rα depicts functionally distinct T cell populations; pro-inflammatory IL-18Rα^hi and cytotoxic IL-18Rα^lo subsets. Our human transcriptomic datasets verified our mouse findings in the settings of human influenza and COVID-19.

In contrast to innate and unconventional T cells that downregulated IL-18Rα following influenza virus infection, CD8 and CD4 T cells substantially upregulated IL-18Rα. This increase in IL-18Rα is linked to T cell activation, a consequence of antigen exposure and further amplified by the increased presence pro-inflammatory stimuli at the site of infection[44,45]. Our key question was to determine the presence of IL-18Rα on immune cell populations in the mouse model

of mild and severe disease, and whether IL-18Rα levels mirrored our data obtained from A/H7N9, COVID-19, RSV and MIS-C patient cohorts. The magnitude of IL-18Rα-positive cells was substantially greater in severe disease, most notably during early timepoints. Specifically, we found that levels of IL-18Rα on CD8 and CD4 T cells from severely infected mice were elevated relative to mild controls, and that high levels of IL-18Rα strongly correlated with exacerbated disease severity (body weight loss), corroborating our previous observations in patient cohorts of life-threatening viral diseases. Future studies utilising IL-18Rα knockout mice would be of interest to understand immune responses in the absence of IL-18Rα^hi immune cells.

The expression of *IL18R1* can be affected by the inflammatory milieu and is upregulated following exposure to pro-inflammatory cytokines[44,45]. Therefore, it is possible that elevated levels of IL-18Rα in severe disease is a consequence of the higher inflammation level, although this does not appear to be linked to plasma IL-18 levels. Excessive inflammation is a common feature of severe respiratory virus infections, including influenza, RSV and SARS-CoV-2[4,46]. Additionally, human infections with highly pathogenic strains of avian-derived influenza, such as A(H5N1) and A(H7N9), are associated with high levels of pro-inflammatory immunomodulators[2]. This hyperinflammation is also reflected in our model of severe influenza, where mice in the severely infected group had markedly elevated levels of cytokines and chemokines in the lungs compared to the mild influenza group.

There is evidence suggesting that CD8 T cells, although critical for recovery, can also drive immunopathology in severe respiratory infection[47]. Indeed, recent studies showed that persistent CXCR3-driven recruitment of cytotoxic CD8 T cells to the lung following the resolution of active infection promoted lung injury in a mouse model of severe influenza. Furthermore, CD8 T cell blockade following viral clearance ameliorated lung vascular injury and led to expedited recovery, highlighting how overexuberant CD8 T cell responses can have deleterious consequences for the host if left unchecked[48]. The association between high levels of IFNγ and large numbers of CD8 T cells was described in the lungs of patients who succumbed to acute respiratory distress (ARDS) following A(H1N1) infection during the 2009 pandemic[49].

Our RNASeq data clearly demonstrated that amongst the most highly upregulated genes in IL-18Rα^hi T cells was *Ifng*, a finding validated in our functionality assays. Schmit et al. demonstrated that CD8 T cell-derived IFNγ exacerbated lung injury in influenza virus infection by promoting the recruitment of CCR2^+ monocytes to the site of infection and supporting their differentiation into a pro-inflammatory, pathologic phenotype[50]. In vitro co-culture studies similarly demonstrated that the combination of IFNγ and TNF derived from human CD8 T cells can lead to bystander lung damage by promoting down-regulation of the epithelial Na,K-ATPase pump on uninfected epithelial cells, thus promoting fluid accumulation into the lung[51]. Likewise, in a murine model of influenza-associated pulmonary aspergillosis, excessive IFNγ production was shown to have resulted in defective Th17-driven immune responses and impaired macrophage function, and that IFNγ ablation promoted improved disease outcomes[52]. Hence,

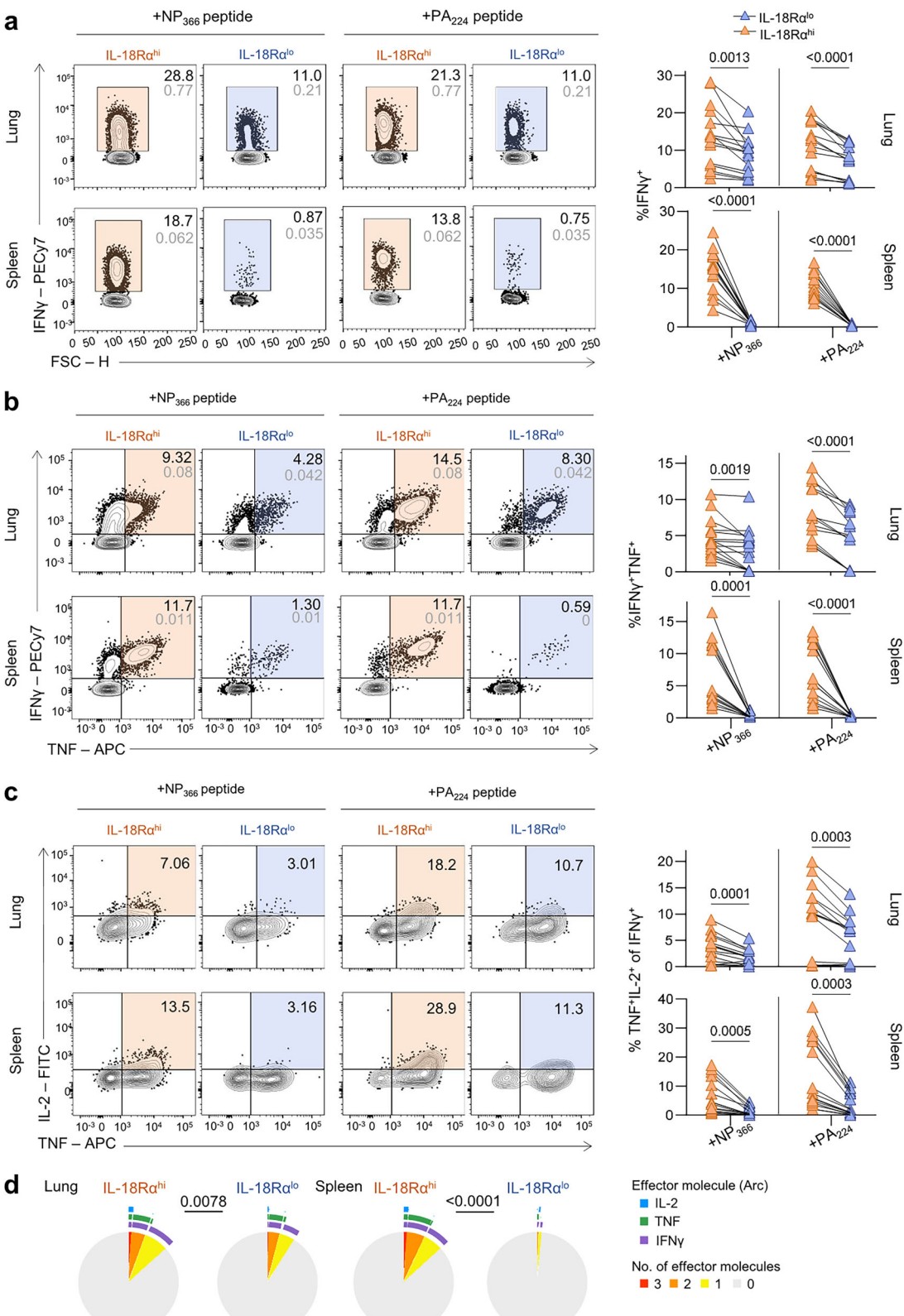

**Fig. 7 | IL-18Rα^hi CD8 T cells are highly polyfunctional. a** Frequency of IFNγ-secreting IL-18Rα^hi and IL-18Rα^lo CD8 T cells in the lung and spleen was analysed ex vivo 10 dpi after peptide stimulation with immunodominant NP$_{366}$ and PA$_{224}$ peptides. **b** Comparison of TNF^+ producers in IFNγ^+IL-18Rα^hi and IL-18Rα^lo CD8 T cells populations. **c** Frequencies of IL-2^+ of IFNγ^+TNF^+CD8^+IL-18Rα^hi and IL-18Rα^lo T cells. Data were analysed using a two-tailed paired t-test (*n* = 15). **d** Polyfunctional profiles (IFNγ, TNF, IL-2) of IL-18Rα^lo and IL-18Rα^hi CD8 T cells after peptide stimulation in the lungs and spleen. Data were analysed via a permutation test. Data shown are from three independent experiments.

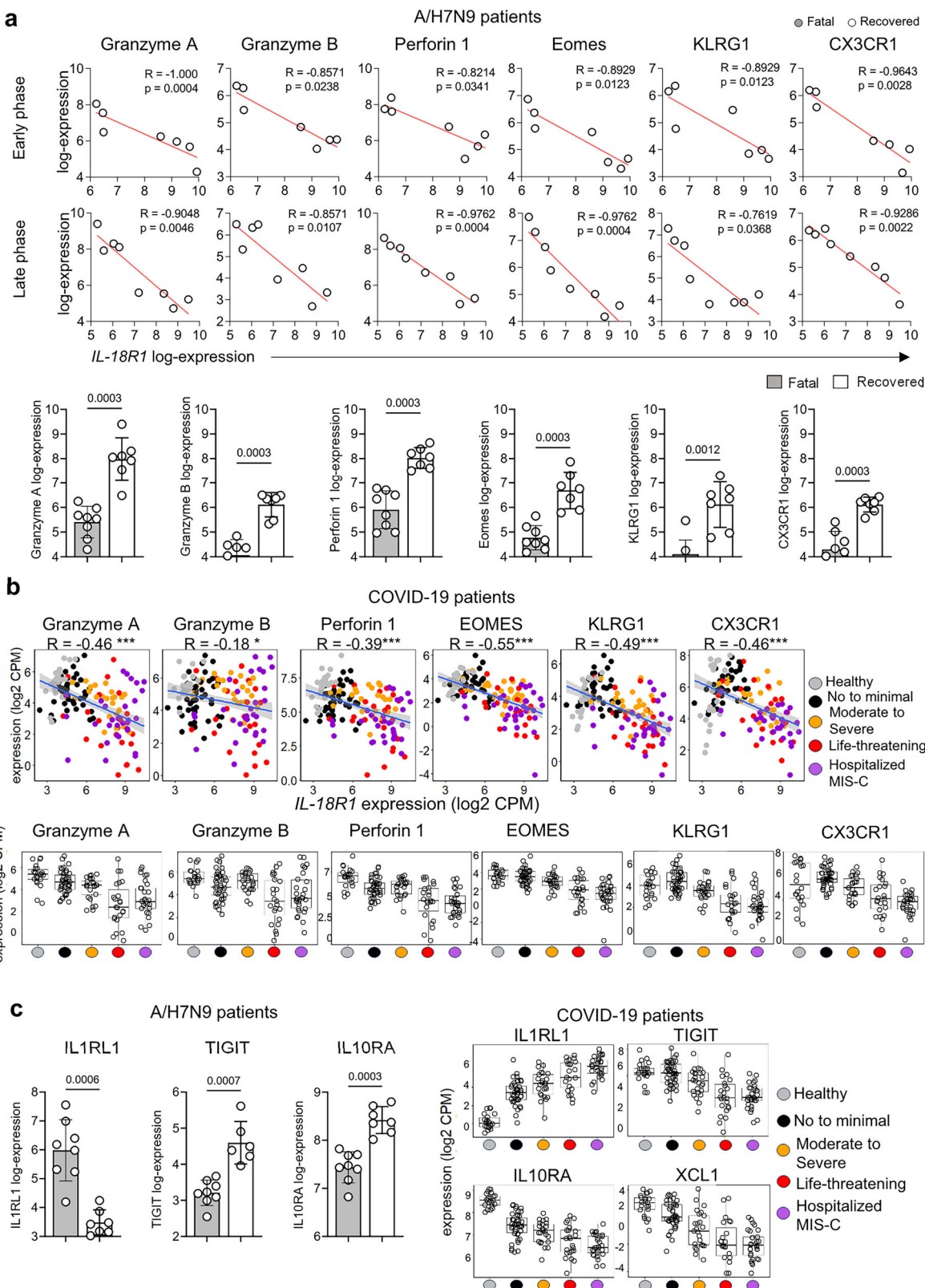

**Fig. 8 | Inverse correlations between IL-18Rα and cytotoxic T cell signatures associated with severe disease outcomes in human A/H7N9 and COVID-19 cohorts.** Correlations between *IL18R1* expression and *GZMA* (Granzyme A)*, GZMB* (Granzyme B)*, PRF1* (Perforin 1)*, EOMES, KLRG1* and *CX3CR1* were analysed in our A/H7N9 and COVID-19 datasets. **a**, **b** Associations between *GZMA* ($p = 8.5e$-09)*, GZMB* ($p = 1.4e$-06)*, PRF1* ($p = 1.4e$-06)*, EOMES* ($p = 2.2e$−16)*, KLRG1* ($p = 6.1e$−10) and *CX3CR1* ($p = 1.5e$−08) expression levels with disease severity was also analysed. **c** Significant correlations with *IL18R1* expression and disease severity for differentially expressed

genes from our OT-I experiment (Fig. 4), namely *IL1R1L1, TIGIT, IL10RA and XCL1*. R = spearman's correlation (two-tailed), where * = $p < 0.05$ l;** = $p < 0.01$; *** = $p < 0.001$. Box plot hinges are the first and third quartiles (the 25th and 75th percentiles). The line between hinges corresponds to the median. The upper whisker extends from the hinge to the largest value no further than 1.5 * IQR from the hinge (where IQR is the inter-quartile range, or distance between the first and third quartiles). The lower whisker extends from the hinge to the smallest value at most 1.5 * IQR of the hinge.

CD8 T cells with high levels of IL-18Rα may contribute to influenza disease severity through pathologic secretion of IFNγ.

Here, we focused predominantly on the role of IL-18Rα expression on CD8 T cells. However, we also found that IL-18Rα was also highly expressed on innate immune cells such as NK cells, and NKT cells. Given the importance of IL-18R/IL-18 signalling in potentiating NK cell activity[53,54], we examined IL-18Rα expression on NK and NK T cells throughout the course of mild and severe influenza disease. Consistent with previous findings showing NK cells constitutively express IL-18R[55] we found that the majority of NK cells expressed IL-18Rα at baseline and expression levels did not substantially change throughout the acute and memory phases of IAV infection. Furthermore, while there were significantly higher frequencies of IL-18Rα+ NK cells in mild disease on day 6 post-infection, these differences were transient and were reversed by day 10. Interestingly, IL-18Rα expression on NK T cells was more dynamic throughout the infection period. Frequencies of IL-18Rα+ NK T cells nearly doubled on day 3 post-infection but plummeted by day 6, before returning to baseline levels on days 10 and 28 post-infection as mice recovered, which may reflect the self-termination of NK T cell activity following primary activation, a phenomenon previously described in microbial infections[56]. A significant correlation between IL-18Rα expression on NK T cells and disease severity was found, but not between IL-18Rα expression on NK cells and disease severity.

Thus, our study demonstrates that IL-18Rα is associated with severe and fatal respiratory disease outcomes, and proposes the use of IL-18Rα as a biomarker for severe disease.

### Limitations of the study

From our study, we cannot conclude that high IL-18Rα expression on its own is sufficient to drive pathogenic T cells, as determinants underpinning disease severity are multifactorial. To address whether IL-18Rα per se can drive pathogenic T cells, future experiments need to be performed in IL-18Rα knockout mice. Future studies should also define in-depth the role of IL-18Rα in CD8 T cell biology.

## Methods

### H7N9, COVID-19 and pARDS patient cohorts

A(H7N9)-infected patients were admitted to the Shanghai Public Health Clinical Centre, and their clinical details have been previously published[2,13]. For our study, whole blood microarray samples from 8 patients[15] were analyzed here according to DEGs for 4 patients who recovered (a73, a134, a20 and a9) and 4 patients with fatal outcomes (a118, a33, a131 and a22) at both early (within 6 days of hospital admission) and late stages of disease (at discharge or 21-22 days post-disease symptom onset). Informed consent was obtained from participants, and the study was approved and conducted under supervision by the SHAPHC Ethics Committee. Blood RNAseq data from the COVID-19 paediatric cohort[15] were analysed for IL-18Rα. Blood from hospitalized patients aged 0-21 years was obtained as part of the Overcoming COVID-19 Study under IRB at Boston Children's Hospital (IRB-P00033157), and the demographics of this cohort have been previously published[15]. Blood samples were collected early after admission. Patient disease severity was grouped according to respiratory involvement comparing 44 children with no to minimal respiratory dysfunction (requiring no major respiratory support other than oxygen or nebulizers), 27 with moderate to severe respiratory dysfunction (with respiratory support; high flow nasal cannula oxygen or non-invasive ventilation) and 24 with life-threatening respiratory failure (requiring invasive mechanical ventilation; with some also requiring extracorporeal membrane oxygenation (1 fatal). Samples from 22 uninfected, healthy individuals recruited at St Jude's Children's Research Hospital (Memphis, TN, USA) as part of the FLU09 cohort[5] were also used as controls in this study. For pARDS analysis, we leveraged existing data from tracheal aspirate samples were obtained

from RSV infected children aged 0-2 recruited at LeBonheur Children's Hospital (Memphis, TN, USA)[21] with 5 experiencing no to mild (n = 5) paediatric ARDS (pARDS), 7 with moderate to severe pARDS and 5 with non-RSV related infection but with moderate to severe pARDS, and 6 control patients who were children without acute lower respiratory tract infection or lung injury. IL18R1 expression was also analysed across time for human challenge models of mild respiratory infections (H1N1 DEE3 n = 477, H3N2 DEE2 n = 355, HRV Duke n = 471, RSV DEE4 n = 420), using previously published datasets[22]. IL18R1 expression in each cohort was analysed from the same patients and disease groupings as was performed for the analysis of differential OLAH expression described in our previous publication (Jia et al, Cell, 2024: PMID 3913778). This included the statistical methods used, models to control for factors and adjustments for multiple comparisons.

### Melbourne cohorts of patients hospitalised with acute respiratory viral diseases

Healthy individuals were recruited via the University of Melbourne. Hospitalized patients with acute respiratory infections were recruited from Austin Health and the Alfred Hospital. All participants provided informed written consent. Experiments conformed to the Declaration of Helsinki Principles and the Australian National Health and Medical Research Council Code of Practice. Ethics approval was provided by the Human Research Ethics Committee of the University of Melbourne (#13344 and #29132), Monash Health (HREC/15/MonH/64), Austin Health (SSA/28204/Austin-2022) and the Alfred Hospital (#280/14). Patient demographics are outlined in Supplementary Tables 1 and 2.

**Antigen-specific tetramer+ CD8 T cell responses.** Biotinylated peptide/HLA-A*02:01 monomer to influenza M1$_{58-66}$ (GILGFVFTL, A2/M1$_{58}$) was generated by the Rossjohn Laboratory (Monash University, Australia), as previously described[57]. HLA-A*02:01 monomer to CMV pp65$_{495-503}$ (NLVPMVATV, A2/pp65$_{495}$) was generated by the McCluskey Laboratory (University of Melbourne, Australia), as previously described[58]. Up to 10 million HLA-A2+ PBMCs from the Melbourne patient cohort were thawed and stained with A2/M1$_{58}$-tetramer-APC for tetramer-associated magnetic enrichment using anti-APC microbeads (Miltenyi Biotec, Bergisch Gladbach, Germany), as previously described[57,59]. Cells were also stained with A2/pp65$_{495}$-tetramer-BV711. Following APC-tetramer enrichment, cell fractions were cell surface stained with anti-human CD71 BV421 (#562995), CD4 BV650 (#563875), CD27 AF700 (#560611), CD38 BV785 (#563964), CD45RA FITC (#555488), CD8 PerCP-Cy5.5 (#565310), CD95 PE-CF594 (#562395), PD-1 PE-Cy7 (#561272), CD14 APC-H7 (#560180), CD19 APC-H7 (#560177) (all from BD Biosciences, Franklin Lakes, NJ, USA), CD3 BV510 (#317332, Biolegend, San Diego, CA, USA), HLA-DR BV605 (#307640, Biolegend), CD218a/IL-18Ra PE (#313808, Biolegend) and L/D NIR (#L10119, Invitrogen, Waltham, MA, USA). Cells were then washed and fixed with 1% PFA before being acquired on a BD LSR Fortessa II. Data were analysed using FlowJo v10 software. Melbourne patient demographics are outlined in Supplementary Tables 1 and 2.

### Mice and influenza virus infection

Both male and female C57BL/6 (H-2$^b$, CD45.2+), and transgenic OT-I (CD45.1+) mice[31] aged 7-12 weeks were bred and maintained under specific pathogen-free conditions at the Melbourne Bioresources Platform at the Peter Doherty Institute, University of Melbourne under a 12 h/12 h light/dark cycle, at 19–22 °C, and 40-70% humidity. To model mild and severe influenza primary infection, mice were intranasally infected with 10$^3$ pfu or 2 × 10$^4$ pfu of A/HKx31 (X31; H3N2), respectively, following light anaesthesia. C57BL/6 mice following transfer of OT-I cells were infected with 10$^5$ pfu of a recombinant X31 strain containing SIINFEKL in the neuraminidase stalk (X31-OVA) at 24 hours post-transfer[32,60]. For challenge experiments, mice were infected with 10$^5$ pfu of PR8 (H1N1) > 28 days after primary infection.

Animals that lost ≥ 25% of their original body weight were humanely killed through carbon dioxide ($CO_2$) asphyxiation. All animal work was conducted in accordance with the Australian National Health and Medical Research Council (NHMRC) Code of Practice for the Care and Use of Animals and approved by the Animal Ethics Experimentation Committee (AEC 20296) at the University of Melbourne.

### Tissue sampling and preparation of single cell suspensions

Lungs, lymph nodes and spleens were harvested and processed into single-cell suspensions. For lungs, perfusion with 5-10 ml of PBS through the right cardiac ventricle was performed prior to removal. Lungs were either enzymatically digested in collagenase III (2 mg/mL Collagenase Type III, Worthington Biochemical Corporation, Cat # LS004182) and DNAse I (10 μg/mL DNAse I, Sigma Aldrich, Cat # 10104159001) at 37 °C for 40 mins before processing, or, homogenised for cytokine analyses. Blood was collected through cardiac puncture following terminal anaesthesia. Samples were coagulated for 5 hrs at RT prior to the collection of serum.

### Staining and flow cytometry

Mouse cells were stained in the dark for 30 mins at 4 °C. The antibodies, including IL-18Rα antibody, are specified in Supplementary Table 3. For identification of influenza-specific CD8 T cells, lymphocytes were stained with $D^bNP_{366-374}$ (ASNENMETM) and $D^bPA_{224-233}$ tetramers (SSLENFRAYV) at RT for 1 hr. Where required, cells were fixed in 1% paraformaldehyde for 20 min at 4 °C prior to acquisition. Viability staining with LIVE/DEAD Fixable Stains (Invitrogen, specified in Supplementary Table 3) was performed at RT for 10 mins prior to staining with antibodies and tetramers. For the analysis of intracellular cytokine levels, cells were first stimulated ex vivo with 1 μM of $NP_{366-374}$ (ASNENMETM) and $PA_{224-233}$ (SSLENFRAYV) peptides (GenScript, Australia) in the presence of 1 μL/mL of Golgi-Plug (BD Biosciences) and 25 U/mL recombinant human IL-2 (Sigma-Aldrich, Cat# 11011456001) for 5 hrs at 37 °C. Cells were then permeabilised using the BD Cytofix/Cytoperm Fixation/Permeabilisation Kit (BD Biosciences, Cat# 554714). To detect cytotoxic molecules and transcription factors Foxp3/Transcription Factor Staining Buffer Set (Invitrogen, Cat#00-5523-00) was used. For in vivo T cell proliferation assays, sorted OT-I cells were stained with 1 μM of Violet Proliferation Dye 450 (VPD450, BD Horizon, Cat# 562158) at $10^7$ cells/mL prior to transfer. Samples were acquired on a BD LSR Fortessa Flow Cytometer (BD).

### In vitro stimulation of cells with IL-12 and IL-18

Bulk lung cells recovered from mice at 6 days after infection with 2 x $10^4$ pfu of X31 were stimulated overnight in the presence of recombinant mouse IL-12 (100 ng; Miltenyi Biotech, Cat # 130-096-707) or IL-18 (100 ng; R&D Systems, Cat # RDS9139IL010) alone or mixed together (100 ng each). The following day, Golgi-Plug (BD Biosciences) was added to each culture for 4 hours at 37 °C prior to surface staining with fluorochrome-conjugated antibodies and intracellular cytokine staining performed.

### Human IL-18Rα expression on whole blood

Whole blood staining for healthy individuals and hospitalized patients was performed as previously described[3,61]. Briefly, blood (200 μl) was directly stained with anti-CD3 PE-CF594, anti-CD8 Per-CP-Cy5.5, anti-CD4 BV650, anti-CD14 AF700 and anti-CD218a/IL-18Rα PE for 30 mins at room temperature in the dark, lysed and fixed before acquisition on LSR Fortessa II (BD). The antibodies are specified in Supplementary Table 3.

### Adoptive cell transfer and isolation of OT-I cells

Lymph nodes were harvested from OT-I/ CD45.1⁺ mice and naïve OT-I cells (identified as $CD8^+CD45.1^+TCRVa2^+CD44^{lo}CD62L^+$) were sorted using FACSAria III (BD, Macquarie Park, NSW, Australia). Unless otherwise stated, $10^5$ naïve OT-I cells were transferred per mouse. $10^6$ OT-I cells were transferred for in vivo proliferation into naïve recipient mice (age and sex-matched). For RNASeq, effector IL-18Rα$^{hi}$ and IL-18Rα$^{lo}$ OT-I cells were sorted from the lungs of X31-OVA infected mice at day 6 post-infection and transferred ($10^5$ cells) into infection-matched recipient mice.

### Bulk RNA sequencing

Effector IL-18Rα$^{hi}$ and IL-18Rα$^{lo}$ OT-I cells were sorted from the lungs of X31-OVA-infected mice. Total RNA was extracted using the RNeasy Micro Plus Kit (QIAGEN, Germany, Cat#74034). RNA concentration and integrity were assessed using an RNA ScreenTape Assay on Agilent TapeStation 4200 (Agilent Technologies, Mulgrave, VIC, Australia). An input of 50 ng of RNA was prepared and indexed for Illumina sequencing using the TruSeq RNA sample Prep Kit (Illumina, Melbourne, Victoria) according to manufacturer's instructions. The library was quantified using the Agilent Tapestation and the Qubit™ DNA assay kit for Qubit 2.0® Fluorometer (Life Technologies, ThermoFisher Scientific). Indexed libraries were prepared and diluted to 500 pM for paired-end (2 × 116 base) sequencing on a NextSeq2000 instrument using the P2 200 cycle kit (Illumina) as specified by the manufacturer. The base calling and quality scoring was performed by the Real Time Analysis (v2.4.6) software. The FASTQ file generation and demultiplexing were performed by the bcl2fastq conversion software (v2.15.0.4).

### Measurement of cytokines

Mouse cytokine and chemokine levels in lung homogenates and serum samples were tested at various timepoints following infection using the LEGENDplex Multi-Analyte Flow Assay Mouse Anti-Virus Response Panel (13-plex) Kit (BioLegend, San Diego, CA, USA, Cat# 740622). Human cytokine and chemokine levels were assessed using the LEGENDplex Human Inflammation Panel 1 Kit (BioLegend, Cat# 740809). Samples were analysed using the LEGENDplex Data Analysis Software Suite (Qognit).

### Statistical analyses. Analysis of flow cytometry data

Unless otherwise stated, flow cytometry data were analysed using FlowJo software version 10.8.1 (FlowJo LLC, BD) and statistical analyses were performed on GraphPad Prism version 9.4.1 (GraphPad Software). Significance was assessed using either a two-tailed Mann-Whitney U-test for unpaired samples, a paired t-test for matched samples or an ordinary one-way ANOVA with Tukey's correction for multiple comparisons. Survival was assessed using a Log-rank (Mantel-Cox) test and differences in body weight loss were compared via area under the curve (AUC). Polyfunctional profiles were performed in SPICE version 6.1 and significance was assessed using a permutation test.

### Analysis of RNA sequencing data

RNAseq data QC and mapping were performed using the bulk RNAseq pipeline of bcbio-nextgen version 1.2.8[62]. Briefly, paired-end reads were mapped to GRCm39 using the STAR aligner with options --outFilterMultimapNmax 10, --limitOutSJcollapsed 2000000, and --sjdbOverhang 115. For downstream analysis quantification was performed using Salmon quant with flags --validateMappings, --seqBias, --gcBias -1 [first pair] -2 [second pair], and --numBootstraps 30. Data quality checks from Samtools[63] version 1.9, Salmon[64] version 1.4.0, STAR[65] version 2.6.1 d and FastQC[66] version 0.11.8 were collated with MultiQC[67] v1.10.1. Analysis was performed on pseudocount data using edgeR version 3.38.4 in R version 4.2.1. Genes were annotated using biomaRt[68] version 2.52.0 and org.Mm.eg.db[69] version 3.15.0. Data were filtered by requiring genes to encode proteins and have counts >10 in one sample and >15 across all samples, and were then scaled for library sizes. A generalised linear model was fitted to paired IL-18Rα$^{hi}$ and IL-18Rα$^{lo}$ samples and differential expression determined by quasi-likelihood

F-test, with false discovery rate <0.05 considered significant. The edgeR functions goana and kegga were used for pathway enrichment analysis, with fry used to perform gene set tests on selected data sets from the literature. Packages used for data visualisation included edgeR, ggplot2[70] version 3.3.6, EnhancedVolcano[71] version 1.14.0 and ComplexHeatmap[72] version 2.12.1. Cytoscape[73] version 3.9.1 was used to display protein-protein interactions among DEG products, which had stringApp[74] version 1.7.1 interaction scores > 0.7. Highly connected clusters were mapped by clusterMaker2[75] version 2.2 using the Markov Cluster Algorithm with an inflation parameter of 2.5.

**Reporting summary**

Further information on research design is available in the Nature Portfolio Reporting Summary linked to this article.

## Data availability

All data generated or analysed during this study are included in this published article (and its supplementary information files) or deposited online. All data are included in the Supplementary information or available from the authors, as are unique reagents used in this Article. The raw numbers for charts and graphs are available in the Source Data. RNAseq data were deposited with the NCBI Sequence Read Archive (#PRJNA1190505). Microarray data from patients infected with A(H7N9) were obtained from GEO accession GSE268303. Bulk RNAseq data from healthy participants, paediatric patients hospitalized with SARS-CoV-2 infection, and paediatric patients hospitalized with MIS-C were obtained from NCBI Sequence Read Archive (SRA) BioProject PRJNA1116218. Single-cell RNAseq data from human tracheal aspirates were obtained from SRA BioProject PRJNA971535. Microarray data from human respiratory virus challenge studies were obtained from GEO accession GSE73072. Source data are provided with this paper.

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

## Acknowledgements

We thank Jill Garlick, Janine Roney, and the research nurses at the Alfred Hospital. This research was funded in part by the National Health and

Medical Research Council of Australia. For purposes of open access, authors applied a CC BY public copyright licence to any Author Accepted Manuscript version arising from this submission. This project has also been funded in part with Federal funds from the National Institute of Allergy and Infectious Diseases, National Institutes of Health, Department of Health and Human Services, Contract #75N93021C00018 (NIAID Centres of Excellence for Influenza Research and Response, CEIRR) to KK, PGT, AGR; Contract #75N93021C00016 to HSS. The authors would like to acknowledge Dr Stephen Wilcox for his expertise and assistance with RNAseq, Prof Chris Chiu for the human challenge datasets and Prof Jamie Rossjohn for the provision of A2/M1$_{58}$ tetramer. We acknowledge the Melbourne Cytometry Platform (The Doherty Institute) for the provision of flow cytometry services. We would like to thank Melbourne Bioresources Platform staff at the Doherty Institute. The work was funded by the NHMRC Leadership Investigator Grant to KK (#2033783) and the Cumming Global Centre for Therapeutic Preparedness grant to KK, LK, BYC, THON and LCR. IJHF was a recipient of the Melbourne Research Scholarship. BYC is a recipient of an NHMRC Ideas Grant (#2001346). XJ was a recipient of the China Scholarship Council-UoM Joint Scholarship. KK is an NHMRC L2 Investigator Fellow (#2033783) and a University of Melbourne Dame Kate Campbell Fellow. J.C.C. and P.G.T. were supported by NIH NIAID R01-AI136514, U01AI150747, and ALSAC at St. Jude. THON and LCR are supported by NHMRC EL Fellowships (#1194036 and #2026357). RCM is supported by a Ruth L. Kirschstein National Research Service Award Individual Postdoctoral Fellowship award (F32AI157296). A.G.R. and P.G.T. are supported by NIH NIAID R01-AI154470.

## Author contributions

K.K. led the study. K.K., L.K., B.Y.C. and T.H.O.N. supervised the study. L.K., K.K., B.Y.C., T.H.O.N., X.J., L.F.A. and A.F.C. designed the experiments. I.J.H.F., L.K., B.Y.C., S.Y.C., X.J., T.H.O.N., R.C.M., L.V.V., L.F.A., R.H., A.T., G.M., B.G. and A.F.C. performed and analysed experiments. H.A.M., D.G., F.L., L.F.A., L.C.R. and S.R. analysed data. M.N.T.S. provided crucial reagents. J.C.C., T.N., J.C., T.F., L.V.V., R.S.T., A.G.R., P.G.T., J.X., Z.W., F.J., E.G., J.T., A.C. and T.C.K. provided clinical samples and analysed data. L.K., K.K., T.H.O.N. and AF.C wrote the manuscript. All authors reviewed and approved the manuscript.

## Competing interests

HAM and BYC consult for Ena Respiratory. AGR received research support from Illumina. PGT is on the SAB of Immunoscape and Cytoagents; consulted for JNJ, received travel support/honoraria from Illumina, 10X Genomics, has patents related to TCR discovery. JCC, PGT have patents related to treating or reducing the severity of viral infections, including SARS-CoV-2. The remaining authors declare no competing interests.

## Additional information

[1]Department of Microbiology and Immunology, The University of Melbourne, at the Peter Doherty Institute for Infection and Immunity, Melbourne, VIC, Australia. [2]Department of Host-Microbe Interactions, St. Jude Children's Research Hospital, Memphis, TN, USA. [3]Center for Infectious Diseases Research, St. Jude Children's Research Hospital, Memphis, TN, USA. [4]Department of Anesthesiology, Critical Care, and Pain Medicine, Boston Children's Hospital and Department of Anaesthesia, Harvard Medical School, Boston, MA, USA. [5]Division of Immunology, Boston Children's Hospital, Harvard Medical School, Boston, MA, USA. [6]Department of Infectious Diseases, Austin Health, Heidelberg, VIC, Australia. [7]School of Medical Sciences and The Kirby Institute, UNSW Sydney, Sydney, NSW, Australia. [8]Department of Pediatrics, The University of Tennessee Health Science Center, Memphis, TN, USA. [9]National Heart and Lung Institute, Imperial College London, London, UK. [10]Department of Infectious Diseases, Peter MacCallum Cancer Centre, Melbourne, VIC, Australia. [11]National Centre for Infections in Cancer, Peter McCallum Cancer Centre, Melbourne, VIC, Australia. [12]Department of Medicine (Austin Health), University of Melbourne, Heidelberg, VIC, Australia. [13]Centre for Antibiotic Allergy and Research, Department of Infectious Diseases, Austin Health, Heidelberg, VIC, Australia. [14]Department of Respiratory Medicine, The Alfred Hospital, Melbourne, VIC, Australia. [15]Department of Medicine, Central Clinical School, The Alfred Hospital, Monash University, Melbourne, VIC, Australia. [16]School of Public Health and Preventive Medicine, Monash University, Melbourne, VIC, Australia. [17]Monash Infectious Diseases, Monash Health and School of Clinical Sciences, Monash University, Clayton, VIC, Australia. [18]Center for Influenza Disease and Emergence Response (CIDER), Athens, GA, USA. [19]Shanghai Public Health Clinical Centre and Institutes of Biomedical Sciences, Key Laboratory of Medical Molecular Virology of Ministry of Education/Health, Shanghai Medical College, Fudan University, Shanghai, China. [20]State Key Laboratory of Respiratory Disease & National Clinical Research Center for Respiratory Disease, Guangzhou Institute of Respiratory Health, the First Affiliated Hospital of Guangzhou Medical University, Guangzhou Medical University, Guangzhou, China. [21]These authors contributed equally: Thi H. O. Nguyen, Brendon Y. Chua, Lukasz Kedzierski, Katherine Kedzierska. ✉e-mail: kkedz@unimelb.edu.au

