## [Transparent Peer Review file · Nature Communications]

High expression of interleukin-18 receptor alpha correlates with severe respiratory viral disease and defines T-cells with reduced cytotoxic signatures

Corresponding Author: Professor Katherine Kedzierska

Version 0:

Reviewer comments:

Reviewer #1

(Remarks to the Author)

In this study, the authors demonstrated that IL-18R α expression of CD8⁺ T cells was associated with the severity of respiratory viral disease. They found that IL18R1 expression of CD8⁺ T cells was significantly elevated in patients with severe disease and correlated with high levels of OLAH, a previously identified biomarker of fatal outcomes. Using a mouse model, they showed that virus-specific IL-18R α hi CD8⁺ T cells exhibit a distinct transcriptional profile compared to IL-18R α lo CD8⁺ T cells, characterized by high IFN-g production but reduced cytotoxic T cell signatures. These findings highlight IL-18R α as a biomarker for severe respiratory viral infections.

Major comments

1. Although the authors identify a strong correlation between IL-18R α expression and severe disease outcomes, further investigation is needed to determine whether IL-18 signaling directly contributes to pathology or is a just correlating marker of disease severity. To clarify its role in pathogenesis, please perform IL-18 blocking experiment in a mouse model.
2. The authors state that high IL18R1 expression of CD8⁺ T cells strongly correlated with OLAH levels. However, a previous study (Cell 2024, 187:4586) reported that epitope-specific CD8⁺ T cells are not affected by OLAH, raising questions about the direct relationship between these two factors. To establish a more definitive correlation, please compare il18r1 expression of CD8⁺ T cells between WT and olah-/- mice. In addition, it might be useful to analyze a correlation between OLAH and IL-18R1 expressions in scRNA-seq.
3. The authors found an association between IL-18R α expression of CD8⁺ T cells and severe disease outcomes, raising the question of whether this phenomenon is exclusive to IL-18 receptor. Please examine the expression levels of other cytokine receptors, such as IL-12 and IL-15, to assess whether this phenomenon is unique to IL-18 receptor.
4. In figure 4i, please examine the proportion of naïve CD8⁺ T cells within the tetramer-negative CD8⁺ T cell population and analyze how this proportion changes across different time points.
5. The authors mentioned that IL-18 can prime T cells via Notch signaling (J Leukoc Biol, 117(1):qjae172). However, in Figure 5g, Notch signaling gene set is enriched in IL-18R α lo cells, not IL-18R α hi cells. Please explain this.
6. In Figure 8, age and CMV serostatus are known to influence the expression of granzyme A, granzyme B, perforin 1, Eomes, KLRG1, and CX3CR1. Please provide age and CMV serostatus of patients. In addition, is there any correlation between age and IL-18R1 expression?
7. In human patients, please examine the expression of IL-18R α in infecting virus-specific MHC multimer⁺ T cells and bystander T cells specific to non-infecting viruses (unrelated virus-specific MHC multimer⁺ T cells).
8. The authors demonstrated distinct characteristics between IL-18R α hi and IL-18R α lo CD8⁺ T cells. While IL-18R α hi cells express proinflammatory cytokines including IFN-g and TNF, IL-18R α lo cells exhibit higher expression of cytotoxic molecules such as granzymes and perforin. Please perform ICS to directly show if IL-18R α hi cells highly produce IFN-g and TNF upon IL-12 and/or IL-18 stimulation. If it is the case, please further discuss about a role of IFN-g produced by IL-18R α hi cells in viral infections.
9. I wonder whether IL-18R α can be used as a predictive biomarker, rather than just disease severity biomarker. Please evaluate this in human patient cohorts.
10. Please conduct adoptive transfer experiment that compare IL-18R α lo and IL-18R α hi cells. I wonder whether IL-18R α lo cells-transferred mice more resist to infection than IL-18R α hi cells-transferred mice.

Minor comment

1. Line 294: Supplementary Fig. 5d -> Supplementary Fig. 3d

Reviewer #2

(Remarks to the Author)

This is an interesting piece of work investigating the inflammatory regulation of severe lower respiratory infection in relation to the IL-18 pathway, specifically the role of IL-18R in severe illness.

The manuscript is well written and the figures well presented. However lack of clarity on the human studies raises questions of sample size, power and how many patients had the full range of assays. If not why not, and were groups comparable by age/disease if not. The manuscript would benefit from a clearer presentation of patient demographics for the human experiments. Apart from sample numbers in the figure legends. For example the numbers of patients, what age groups, what cohorts and their severity/outcomes. Much of the data uses samples from previously published studies so for ease of understanding. it would be useful to have a section in the results alongside the experimental flow on the patient cohorts and their demographics. Also clarity on how data is comparable from one set to another in terms having been taken from the same patients.

The investigators should reframe their manuscript's novelty to incorporate existing published work by other researchers who have investigated this cohort (COVID/MIS-C/Severe viral illness) in children and adults and demonstrated upregulation of the IL-18 signalling pathway. They should include discussion of and their findings and how it integrates into the context of these other works.

Reviewer #3

(Remarks to the Author)

The paper was well done, although some points can be improved for the readability of the paper.

Firstly: in the introduction the role of IL-18 was explained in the context of inflammation mediated by viral infection. Although it is not clearly reported the dynamics and the biology which explain the link between IL-18 and OLAH.

Moreover, in the final paragraph of the introduction (page 4 line 113) it is too much long the explanation of the results. I suggest to the authors to modulate this paragraph and move part of these in the discussion and better explain the hypothesis based on recent literature data.

Concerning materials and methods, it could be of interest to add a kind of "flow chart" with the selection of patients and the enrollment criteria which better explain the cohort description.

Moreover, in my opinion it could be of interest also to perform a statistical analysis which explain that other variables (such as the comorbidities of the patients) do not affect the results related to "inflammatory status of the patients" and relative IL18R alpha expression.

In my opinion it could be also interesting to deeply explain the results regarding the expression of IL-18 in NKT and NK cells. Due to the extreme correlation between innate and adaptive immunity in these kinds of patients and especially in the correlation with OLAH, please better explain in the discussion these data. I also suggest performing other analysis regarding these cellular clusters.

The data regarding CD38, PD-1 and HLA-DR. In my opinion the role of these markers in the infection are still debates. This paper opens new frontiers in this field. Although for the readability of the paper I suggest adding a paragraph in the introduction which better explains why the authors choose to explore the exhaustion state of the cells.

Version 1:

Reviewer comments:

Reviewer #1

(Remarks to the Author)

The authors have revised the manuscript in response to reviewers' comments.

However, I have further questions and comments.

1. Reviewer #1's comment #2

I appreciate the authors' effort to address this comment and understand the result. If this is the case, I would like to recommend deleting the OLAH-related parts from this manuscript, including Figures 1C, 1F, 3C, 3F, and related contents. You may instead discuss OLAH as an independent biomarker in the Discussion. The story of IL-18Ra expression on CD8 T cells is sufficient to be the main topic of this manuscript, even without the data on its correlation with OLAH. The current manuscript places too much emphasis on the correlation between IL-18Ra and OLAH, which could mislead readers to assume a causal relationship between OLAH and IL-18Ra expression.

2. Reviewer #1's comment #8

I understand the authors' response. However, if this is the case, many readers will likely wonder: if IL-18Ra-hi CD8 T cells produce IFN-g very well, is IFN-g beneficial or detrimental to the host? Please add discussion on the role of IFN-g produced by IL-18Ra-hi CD8 T cells during viral infections in the Discussion section.

Reviewer #2

(Remarks to the Author)

The manuscript is much improved and I appreciate the clarity in methods and discussion in the revised manuscript. I have no further comments

Reviewer #3

(Remarks to the Author)

The paper has been significantly improved, and the authors have followed the reviewers' suggestions. In my opinion the paper can be accepted for publication.

Point-by-point responses to Reviewers' comments

We thank the Reviewers for their comments and insightful suggestions, which have allowed us to further improve the manuscript. We would also like to thank the Editors for the opportunity to resubmit the revised version to *Nature Communications*.

Reviewer #1:

In this study, the authors demonstrated that IL-18R α expression of CD8⁺ T cells was associated with the severity of respiratory viral disease. They found that IL18R1 expression of CD8⁺ T cells was significantly elevated in patients with severe disease and correlated with high levels of OLAH, a previously identified biomarker of fatal outcomes. Using a mouse model, they showed that virus-specific IL-18R α hi CD8⁺ T cells exhibit a distinct transcriptional profile compared to IL-18R α lo CD8⁺ T cells, characterized by high IFN-g production but reduced cytotoxic T cell signatures. These findings highlight IL-18R α as a biomarker for severe respiratory viral infections.

We thank the Reviewer for their supportive comments.

Major comments

1. Although the authors identify a strong correlation between IL-18R α expression and severe disease outcomes, further investigation is needed to determine whether IL-18 signaling directly contributes to pathology or is a just correlating marker of disease severity. To clarify its role in pathogenesis, please perform IL-18 blocking experiment in a mouse model.

We thank the Reviewer for this important question. Following the Reviewer's comment, we have performed the recommended experiments.

As per previously published literature (Robinson et al, JCI Insight 2018, PMID 29618653), we administered to the mice 200 μ g of rat anti-mouse IL-18 (clone YIGIF74-1G7) or IgG2a isotype control (2A3) antibody (both from Bio X Cell InVivoMAb, New Hampshire, USA) at day 3 after influenza virus infection via the intra-peritoneal route. We performed 2 experiments.

Experiment 1:

We monitored mice for their body weight loss until day 7. We also collected lungs on day 7 after influenza virus infection and assayed viral load titres via the plaque assay as well as cytokine and chemokine mediators from lung homogenate.

Our data, presented in Rebuttal Fig 1 below, show that while the lung viral load was significantly reduced in the anti-IL-18 animal group on d7 after infection (Rebuttal Fig 1a), there was a trend towards both increased body weight loss (Rebuttal Fig 1b) and increased inflammation in the lungs following IL-18 blockade (Rebuttal Fig 1c), indicating exacerbated disease severity. This experiment thus suggests that blocking IL-18 does not have any clear effects on preventing influenza disease severity.

Rebuttal Figure 1. Mice infected with 2×10^4 pfu of X31 were inoculated with 200 μg of rat anti-mouse IL-18 (clone YIGIF74-1G7) or IgG2a isotype control (2A3) antibody (both from Bio X Cell InVivoMAb, New Hampshire, USA) at day 3 after infection via the intra-peritoneal route. We monitored mice for body weight loss until day 7. We also collected lungs on day 7 after influenza virus infection, and assayed for viral load titres via an MDCK plaque assay as well as cytokine and chemokine mediators.

Experiment 2:

In the second experiment, performed as above, we monitored mice for 10 days to further assess the impact of IL-18 blockade on disease severity depicted by the body weight loss. However, in this experiment, 2 of the mice in the anti-IL-18 antibody treatment group lost >25% body weight and needed to be euthanised at this point (day 6, Rebuttal Fig 2a). These mice had excessive lung inflammation, especially with respect to IL-6 production, indicating they potentially died of the cytokine storm (Rebuttal Fig 2b).

Rebuttal Fig 2. Disease severity in influenza virus-infected mice treated with the anti-IL-18 blocking antibody. Mice were infected with 2×10^4 PFU of X31 influenza virus and monitored for 10 days for disease severity, as depicted by their body weight loss. On day 3 after infection, mice were treated i.p. with either the isotype control or anti-IL-18 antibody (200 μ g/mouse). (a) Disease severity was assessed by measuring the body weight loss. An arrow on d6 depicts the time-point when 2 mice from the anti-IL-18 treatment group were culled because of >25% body weight loss; (b) Lung cytokine and chemokine levels were determined for 2 mice (M1 & M2) which succumbed to the influenza viral infection on day 6 after infection.

Thus, overall, the results from our 2 experiments show that the protective (viral load) versus pathologic (inflammation and body weight loss) effects of IL-18 signalling in influenza virus infection are complex and need to be understood in depth to decipher the role of IL-18 in finetuning the balance between recovery versus fatal disease outcomes. As such, we feel that future studies are needed before the data can be shared with the readers.

Furthermore, it is important to note that these IL-18 blocking experiments in mice support our findings from the human patient cohort, included in our manuscript as Supplementary Figure 2. In human settings, when we analysed plasma IL-18 cytokine levels in our human cohort of patients hospitalised with seasonal influenza viruses, we demonstrated a lack of correlation between IL-18R α and IL-18 in our seasonal influenza disease cohort as well as in our H7N9 dataset.

Our manuscript focuses on the strong link between IL-18R α (but not IL-18) with disease severity and OLAH.

2. The authors state that high IL18R1 expression of CD8+ T cells strongly correlated with OLAH levels. However, a previous study (Cell 2024, 187:4586) reported that epitope-specific CD8+ T cells are not affected by OLAH, raising questions about the direct relationship between these two factors. To establish a more definitive correlation, please compare il18r1 expression of CD8+ T cells between WT and olah-/- mice.

Following the Reviewer's recommendation, we have investigated IL-18R α expression on CD8+ T cells in *olah*^{-/-} mice on day 3 and day 6 following influenza virus infection. We have performed 3 independent experiments for day 3 (n=17 for WT, n=13 for *olah*^{-/-} mice) and 2 independent experiments for day 6 (n=10 for WT, n=10 for *olah*^{-/-} mice). Overall, our data showed no significant changes in either the frequency or the number of IL-18R α ^{hi} total CD8+ T cells in the absence of OLAH. Similarly, the number of IL-18R α ^{hi} effector CD8+ T cells remained unchanged in *olah*^{-/-} mice following influenza virus infections. Our results thus indicate that OLAH expression per se does not directly impact IL-18R α expression on CD8+ T cells in lungs of influenza-infected mice. This implicates that the high expression levels of both OLAH and IL-18R α , strongly linked with severe respiratory viral diseases in our human data (Fig. 1 and Fig. 2), are mediated via an independent mechanism.

We have now included these data in the revised version of the manuscript (Results; page 10):

“To understand any potential link between IL-18R α expression and OLAH, we investigated IL-18R α surface expression on CD8 $^+$ T cells in *olah* $^{-/-}$ mice on day 3 and day 6 following influenza virus infection. We have performed 3 independent experiments for day 3 (n=17 for WT, n=13 for *olah* $^{-/-}$ mice) and 2 independent experiments for day 6 (n=10 for WT, n=10 for *olah* $^{-/-}$ mice). Overall, our data demonstrated no significant changes in either the frequency or the number of IL-18R α $^{\text{hi}}$ total CD8 $^+$ T cells in the absence of OLAH (Supplementary Fig. 8). Similarly, the number of IL-18R α $^{\text{hi}}$ effector CD8 $^+$ T cells remained unchanged in *olah* $^{-/-}$ mice following influenza virus infections. Our results thus indicate that OLAH expression per se does not directly impact IL-18R α expression on CD8 $^+$ T cells in lungs of influenza-infected mice. This implies that high expression levels of both *olah* and IL-18R α , strongly linked with severe respiratory viral diseases in our human data (Fig. 1 and Fig. 2), are mediated via an independent mechanism.”

Supplementary Fig. 8. IL-18R α expression on CD8 $^+$ T cells remains unchanged in influenza-infected *olah* $^{-/-}$ mice. IL-18R α staining in the lungs of WT and *olah* $^{-/-}$ mice following IAV infection. Frequencies and absolute numbers of IL-18R α $^{\text{hi}}$ CD8 $^+$ T cells and IL-18R α $^{\text{hi}}$ CD8 $^+$ effector T cells (CD62L $^{\text{lo}}$ CD44 $^{\text{hi}}$) were assessed on 3 and 6 following influenza virus infection. Data plotted are from 3 independent experiments for day 3 (n=17 for WT, n=13 for *olah* $^{-/-}$) and 2 independent experiments for day 6 (n=10 for WT, n=10 for OLAH $^{-/-}$). Bars represent mean \pm SEM, statistically significant differences between groups were determined using unpaired Student’s t-test.

3. The authors found an association between IL-18R α expression of CD8 $^+$ T cells and severe disease outcomes, raising the question of whether this phenomenon is exclusive to IL-18 receptor. Please examine the expression levels of other cytokine receptors, such as IL-12 and IL-15, to assess whether this phenomenon is unique to IL-18 receptor.

Following the Reviewer’s comment, we have now performed additional analyses of the transcript levels for IL-12 and IL-15 receptors in our human cohorts. As shown in Supplementary Figure 1, we found no significant differences in *IL15RA* and *IL12RB1* levels between fatal and recovered A/H7N9 patients across both early and late time points (Supplementary Fig. 1a). Both receptors showed no correlation with OLAH at

the early time points. At the late time points, however, there was an inverse correlation between decreased *IL15RA* and *IL12RB1* with higher *OLAH* (Supplementary Fig. 1a).

Similarly, *IL15RA* levels remained unchanged across disease severities in hospitalized SARS-CoV-2 and MIS-C patients compared to healthy individuals (Supplementary Fig. 1b), in contrast to *IL18R1* expression (Fig 1e). However, *IL12RB1* and *IL12RB2* transcript levels as well as those encoding for the cytokines, IL15 and IL12 (*IL15* and *IL12A*, respectively) decreased in life-threatening COVID-19 and MIS-C (Supplementary Fig. 1c).

Supplementary Figure 1. Transcript expression levels of *IL15RA* and *IL12RB1* in life-threatening H7N9 and COVID-19. *IL15RA* and *IL12RB1* transcript expression levels in (a) A/H7N9 patients from fatal and recovered groups. (b) healthy individuals and patients with SARS-CoV-2 infection across disease severity. (c) *IL15* and *IL12A* transcript levels in COVID-19. Boxplots show transcriptional expression of genes of interest as a function of disease severity. P-values were obtained from a model that controlled for the effects of days since symptoms onset, sex, whether a patient was previously healthy, steroid administration prior to sampling, bacterial co-infection, age, race, and ethnicity, and were adjusted for multiple comparisons.

We have included these data in our revised version of the manuscript (Supplementary Fig. 1) and described as follows (Results, page 6):

“In contrast to increased *IL-18R1* expression, we found no significant differences in *IL15RA* and *IL12RB1* levels between fatal and recovered A/H7N9 patients across both early and late time points (Supplementary Fig. 1a). Both receptors showed no correlation with *OLAH* at the early time points. At the late time points, however, there

was an inverse correlation between decreased *IL15RA* and *IL12RB1* with higher OLAH (Supplementary Fig. 1a). Similarly, *IL15RA* levels remained unchanged across disease severities in hospitalized SARS-CoV-2 and MIS-C patients compared to healthy individuals (Supplementary Fig. 1b), in contrast to *IL18R1* expression (Fig 1e). However, *IL12RB1* and *IL12RB2* transcript levels as well as those encoding for the cytokines, IL15 and IL12 (*IL15* and *IL12A*, respectively) decreased in life-threatening COVID-19 and MIS-C (Supplementary Fig. 1c).”

4. In figure 4i, please examine the proportion of naïve CD8+ T cells within the tetramer-negative CD8+ T cell population and analyze how this proportion changes across different time points.

We have performed the analyses as recommended by the Reviewer. We have included the above findings in Results (page 11) and Fig 4j:

“We have analysed the proportion of naïve ($CD62L^{hi}CD44^{lo}$) CD8+ T cells within the tetramer-negative population on days 10 and 28 post primary infection, and day 8 following secondary infection (Fig.4j). Compared to day 10 and day 8 (2°) post-infection, there is a significantly higher proportion of naïve tetramer-negative CD8+ T cells on day 28. These observations are consistent with the fact that a proportion of the tetramer-negative CD8+ T cells become activated and transition into $CD44^{hi}$ effectors during the acute phases of influenza virus infection, resulting in fewer naïve CD8+ T cells compared to a memory (day 28) timepoint, during which there is no longer active inflammation or viral replication.”

Fig.4j. Frequency of naïve cells within D^bNP_{366} and D^bPA_{224} -tetramer negative CD8+T cells.

5. The authors mentioned that IL-18 can prime T cells via Notch signaling (J Leukoc Biol, 117(1):qiae172). However, in Figure 5g, Notch signaling gene set is enriched in IL-18Ralo cells, not IL-18Rahi cells. Please explain this.

We thank the Reviewer for this question.

The J Leukoc Biol manuscript by Wen Li *et al* used a cancer model to demonstrate that IL-18 can prime innate-like CD44^{hi}CD122^{hi}CXCR3^{hi}CD62L^{hi}CD8⁺ T cells with an antigen-inexperienced memory phenotype into effector-like populations in a Notch signalling-dependent manner.

In our study, we assessed gene transcription profiles in influenza antigen-specific IL-18R^{lo} and IL-18R^{hi} CD8⁺ T cells, hence in our study, the antigen-specific CD8⁺ T cells are different to innate-like antigen-inexperienced CD8⁺ T cells used in Wen Li *et al* publication.

We believe that the findings in Wen Li *et al* could be analogous to the higher expression of Notch-related genes observed in our IL-18R^{lo} CD8⁺ T cell population, likely reflecting a less differentiated (naïve-like) population but more receptive to activation involving this signalling pathway compared to the IL-18^{hi}CD8⁺ T cell population.

However, for clarity, following the Reviewer's comment, we deleted the Wen Li *et al* reference not to confuse the reader.

6. In Figure 8, age and CMV serostatus are known to influence the expression of granzyme A, granzyme B, perforin 1, Eomes, KLRG1, and CX3CR1. Please provide age and CMV serostatus of patients. In addition, is there any correlation between age and IL-18R1 expression?

Following the Reviewer's comment, we investigated the effect of age on the expression of granzyme A, granzyme B, perforin 1, Eomes, KLRG1, CX3CR1 and IL-18R1 and found no significant differences (Supplementary Fig. 12).

We included these data in Results (page 16):

“Our analysis of *GZMA*, *GZMB*, *PRF1*, *EOMES*, *KLRG1*, *CXCR1* and IL-18R1 expression levels stratified according to these age ranges showed no clear correlation between age and expression of any of these genes (Supplementary Fig. 12).”

Supplementary Fig. 12. *GZMA*, *GZMB*, *PRF1*, *EOMES*, *KLRG1*, *CXCR1* and *IL-18R1* expression levels stratified according to age ranges. (a) Boxplots showing transcriptional expression of genes as a function of age category. (b) *IL18R1* expression did not vary significantly as a function of age category, sex, or BMI (R_s : Spearman's rank correlation).

CMV serostatus of patients is unavailable.

7. In human patients, please examine the expression of IL-18R α in infecting virus-specific MHC multimer+ T cells and bystander T cells specific to non-infecting viruses (unrelated virus-specific MHC multimer+ T cells).

We thank the Reviewer for this suggestion. We have now performed the experiments in 10 HLA-A*02:01-expressing patients hospitalised with influenza. We indeed found statistically increased surface expression of IL-18R α on influenza tetramer-specific CD8⁺ T cells in human settings, in agreement with our mouse data (Fig. 4i). These results are described in Results, Methods and included in the revised version of our manuscript as Figure 2c (also shown below).

Results (page 7):

“To determine whether IL-18R α expression was further increased in influenza-specific CD8⁺ T cells compared to the bulk CD8⁺ T cell population in hospitalized influenza-A patients, we performed tetramer-associated magnetic enrichment on a subset of HLA-A2⁺ patients (n=10) to enrich for influenza-specific CD8⁺ T cells recognizing the immunodominant A2/M1₅₈ epitope (Supplementary Fig. 2b). Indeed, influenza-specific CD8⁺ T cells had higher IL-18R α expression (mean 77.0%) compared to bulk CD8⁺ T

cells (59.3%) (Fig. 2c, $P = 0.0068$), whereas there was no difference in IL-18R α expression for unrelated CMV-specific CD8 $^+$ T cells (69.9%).”

Methods (page 22):

“Antigen-specific tetramer $^+$ CD8 $^+$ T cell responses

Biotinylated peptide/HLA-A*02:01 monomer to influenza M1₅₈₋₆₆ (GILGFVFTL, A2/M1₅₈) was generated by the Rossjohn Laboratory (Monash University, Australia), as previously described (Louise Immunity 2022 #35750048). HLA-A*02:01 monomer to CMV pp65₄₉₅₋₅₀₃ (NLVPMVATV, A2/pp65₄₉₅) was generated by the McCluskey Laboratory (University of Melbourne, Australia), as previously described (Nguyen JI 2014 #24778446). Up to 10 million HLA-A2 $^+$ PBMCs from Melbourne patient cohort were thawed and stained with A2/M1₅₈-tetramer-APC for tetramer-associated magnetic enrichment using anti-APC microbeads (Miltenyi Biotec, Bergisch Gladbach, Germany), as previously described (Rowntree *et al*, Immunity 2022, Rowntree *et al*, PNAS 2024). Cells were also stained with A2/pp65₄₉₅-tetramer-BV711. Following APC-tetramer enrichment, cell fractions were cell surface stained with anti-human CD71 BV421 (#562995), CD4 BV650 (#563875), CD27 AF700 (#560611), CD38 BV785 (#563964), CD45RA FITC (#555488), CD8 PerCP-Cy5.5 (#565310), CD95 PE-CF594 (#562395), PD-1 PE-Cy7 (#561272), CD14 APC-H7 (#560180), CD19 APC-H7 (#560177) (all from BD Biosciences, Franklin Lakes, NJ, USA), CD3 BV510 (#317332, Biolegend, San Diego, CA, USA), HLA-DR BV605 (#307640, Biolegend), CD218a/IL-18Ra PE (#313808, Biolegend) and L/D NIR (#L10119, Invitrogen, Waltham, MA, USA). Cells were then washed and fixed with 1% PFA before acquiring on a BD LSR Fortessa II. Data were analyzed using FlowJo v10 software. Melbourne patient demographics are outlined in Supplementary Table 1.”

Fig. 2c. High surface expression of IL-18R α on human tetramer $^+$ CD8 $^+$ T cells during severe respiratory disease. (c) Representative FACS plots of surface IL-18R α expression on unenriched CD8 $^+$ T cells, tetramer-enriched influenza-specific A2/M1₅₈ $^+$ CD8 $^+$ T cells and unenriched CMV-specific A2/pp65₄₉₅ $^+$ CD8 $^+$ T cells in influenza A patients. Graphed IL-18R α expression in 10 HLA-A2 $^+$ patients hospitalized

with influenza A at acute timepoints. Columns indicate mean \pm SD. Statistical significance was analysed using Wilcoxon matched-pairs signed rank test. Patient demographics are outlined in Supplementary Table 1.

8. The authors demonstrated distinct characteristics between IL-18R^{hi} and IL-18R^{lo} CD8⁺ T cells. While IL-18R^{hi} cells express proinflammatory cytokines including IFN- γ and TNF, IL-18R^{lo} cells exhibit higher expression of cytotoxic molecules such as granzymes and perforin. Please perform ICS to directly show if IL-18R^{hi} cells highly produce IFN- γ and TNF upon IL-12 and/or IL-18 stimulation. If it is the case, please further discuss about a role of IFN- γ produced by IL-18R^{hi} cells in viral infections.

We thank the Reviewer for this comment. Following the Reviewer’s recommendation, we have performed IFN γ intracellular cytokine staining assay of IL-18R α^{lo} and IL-18R α^{hi} CD8⁺ T cells from lungs on day 6 following influenza virus infection. We stimulated lung cells overnight with either IL-12, IL-18 or IL-12+IL-18. Unstimulated negative control and PMA/I stimulated positive control were included.

Our data show (Results; page 13):

“In agreement with our transcriptomics data, we found markedly increased capacity of influenza-infected lung IL-18R α^{hi} CD8⁺ T cells to produce IFN γ following stimulation with IL-12 and IL-18 overnight, while there was a clear lack of IFN γ production by lung IL-18R α^{lo} CD8⁺ T cells following influenza virus infection (Figure 5h).”

Figure 5h: IFN γ intracellular cytokine staining of IL-18R α^{lo} and IL-18R α^{hi} CD8⁺ T cells from lungs on day 6 following influenza virus infection. Lymphocytes were stimulated overnight with either IL-12, IL-18 or IL-12 + IL-18. Unstimulated negative control and PMA/I stimulated positive control were included. Top panel: IFN γ production by IL-18R α^{hi} CD8⁺ T cells; bottom panel: IFN γ production by IL-18R α^{lo} CD8⁺ T cells. Bars represent mean \pm SD, statistically

significant differences between groups were determined using one-way ANOVA Kruskal-Wallis test with Dunn's multiple comparison test, **P<0.01. ns: not significant.

Methods (page 24):

"In vitro stimulation of cells with IL-12 and IL-18

Bulk lung cells recovered from mice at 6 days after infection with 2×10^4 pfu of X31 were stimulated overnight in the presence of recombinant mouse IL-12 (100ng; Miltenyi Biotech) or IL-18 (100ng; R&D Systems) alone or mixed together (100ng each). The following day, Golgi-Plug was added to each culture for 4 hours at 37°C prior to surface staining with fluorochrome-conjugated antibodies and intracellular cytokine staining performed."

It is important to note that no TNF or IL-2 was detected in our assay.

9. I wonder whether IL-18R α can be used as a predictive biomarker, rather than just disease severity biomarker. Please evaluate this in human patient cohorts.

Yes, we are fully agreeing with the Reviewer. Indeed, our current studies, supported by the Cumming Global Centre for Pandemic Therapeutics funds, are aimed to develop a rapid screening clinical blood test using both OLAH and IL-18R α as predictive markers of disease severity on hospital admission to inform clinicians on potential life-threatening disease outcomes. We have stated this in Discussion (page 20):

"Our current studies aim at developing a clinical screening test for both OLAH and IL-18R α to predict disease severity on hospital admission."

10. Please conduct adoptive transfer experiment that compare IL-18R α ^{lo} and IL-18R α ^{hi} cells. I wonder whether IL-18R α ^{lo} cells-transferred mice more resist to infection than IL-18R α ^{hi} cells-transferred mice.

We thank the Reviewer for this comment. We agree with the Reviewer that showing that IL-18R α ^{lo} CD8⁺ T cells-transferred mice are more resistant to infection than IL-18R α ^{hi} CD8⁺ T cells-transferred mice would be of great interest. However, as per our previous experience (and decades of experimental work with transfer studies), immunity to influenza viruses is rapid and multifactorial. Removing/adding one component does not necessarily equate to improved/worsened disease outcomes. We believe that to show the protective effect, we would require the IL-18R α knock out mice. However, we do not have at the present time access to IL-18R α knock out mice.

Following Reviewer's comment, we added these experiments as future studies in our manuscript (Discussion, page 18):

"Future studies utilising IL-18R α knock out mice would be of interest to understand immune responses in the absence of IL-18R α ^{hi} immune cells."

Minor comment

1. Line 294: Supplementary Fig. 5d -> Supplementary Fig. 3d

We thank the Reviewer. We have corrected this Supplementary Figure.

Reviewer #2:

This is an interesting piece of work investigating the inflammatory regulation of severe lower respiratory infection in relation to the IL-18 pathway, specifically the role of IL-18R in severe illness. The manuscript is well written and the figures well presented.

We thank the Reviewer for their supportive comments.

However lack of clarity on the human studies raises questions of sample size, power and how many patients had the full range of assays. If not why not, and were groups comparable by age/disease if not. The manuscript would benefit from a clearer presentation of patient demographics for the human experiments. Apart from sample numbers in the figure legends. For example the numbers of patients, what age groups, what cohorts and their severity/outcomes. Much of the data uses samples from previously published studies so for ease of understanding. it would be useful to have a section in the results alongside the experimental flow on the patient cohorts and their demographics.

We thank the Reviewer for this question. To clarify our human cohorts, in the modified version of our manuscript we have now included:

- (1) A clear summary of the previously published human cohorts in Figure 1a, as below:

a

Cohort	Country	n	Age range	Analysis
A/H7N9 (Ref 15)	China	Fatal (4) Recovered (4)	53-88	Microarray (blood)
SARS-CoV-2 (Ref 15)	USA	Healthy (22) Infected (95) MIS-C (30)	0-21	Bulk RNASeq (blood)
pARDs (Ref 15, 21)	USA	RSV-infected (12) Non-RSV infection (5) Uninfected (6)	0-2	scRNASeq (tracheal aspirates)
Healthy (Ref 5)	USA	Uninfected (22)		Bulk RNASeq (blood)

Figure 1a. Summary of previously published disease cohorts, group sample sizes, age ranges and data used for IL-18R α analyses.

In addition, we have amended the text to provide greater clarity on which cohort is discussed as well as more information on the demographics or where they have been published.

Results (page 4):

“Here, in the same A(H7N9) cohort, our analysis of differentially expressed genes identified the *IL18R1* gene, encoding for IL-18R α receptor (CD218a; IL-18R α), as one of the three key genes highly expressed in fatal H7N9 patients early after hospital admission (within 6 days of hospital admission), alongside *OLAH* and *FLT3* (Fig. 1b)

“To determine whether the observed association between *IL18R1* levels and acute A(H7N9) disease severity could also be recapitulated in other respiratory diseases, we assessed *IL18R1* gene expression in a second disease cohort of patients hospitalized with acute SARS-CoV-2 infection and MIS-C (multisystem inflammatory syndrome in children)(Fig. 1a).”

Results (page 5):

“In comparison to samples from a cohort of healthy individuals that were used as controls (Fig 1a), *IL18R1* expression was significantly elevated in patients hospitalized with COVID-19 with no to minimal respiratory dysfunction (P=0.019).”

“In our third unrelated disease cohort²¹, single-cell RNA sequencing (scRNAseq) data were obtained from tracheal aspirates collected from children hospitalized with acute respiratory failure requiring endotracheal intubation, stemming from lower respiratory tract infection (LRTI)(Fig 1a).”

Materials and Methods (page 21):

“H7N9, COVID-19 and pARDS patient cohorts. A/H7N9-infected patients were admitted to the Shanghai Public Health Clinical Center between March and August 2013 and their clinical details have been previously published^{2, 13}. For our study, whole blood microarray samples from 8 patients¹⁵ were analyzed here according to DEGs for 4 patients who recovered (a73, a134, a20 and a9) and 4 patients with fatal outcomes (a118, a33, a131 and a22) at both early (within 6 days of hospital admission) and late stages of disease (at discharge or 21-22 days post-disease symptom onset). Informed consent was obtained from participants, and the study was approved and conducted under supervision by the SHAPHC Ethics Committee.”

“Blood from hospitalized patients aged 0-21 years was obtained as part of the Overcoming COVID-19 Study under IRB at Boston Children’s Hospital (IRB-P00033157) and the demographics of this cohort have been previously published¹⁵.”

Materials and Methods (page 21):

“Samples from 22 uninfected, healthy individuals recruited at St Jude’s Children’s Research Hospital (Memphis, TN, USA) as part of the FLU09 cohort⁵ were also used as controls in this study.”

“For pARDS analysis, we leveraged existing data from tracheal aspirate samples obtained from RSV infected children aged 0-2 recruited at LeBonheur Children’s Hospital (Memphis, TN, USA)²¹ with 5 experiencing no to mild (n=5) pediatric ARDS (pARDS), 7 with moderate to severe pARDS and 5 with non-RSV related infection but with moderate to severe pARDS, and 6 control patients who were children without acute lower respiratory tract infection or lung injury.”

- (2) A detailed Supplementary Table 1 outlining demographics of the Melbourne cohort of patients hospitalised with respiratory viral diseases used in Figure 2 and Figure 3, as below:

Supplementary Table 1. Demographics of the acute viral respiratory hospitalised patient cohort in Melbourne.

Cohort Summary	Healthy	Patients
----------------	---------	----------

Number of individuals, n	17	43
Age, median (range)	22 (19-52)	68 (26-89)
Female, n (%)	14 (82%)	20 (47%)
Days post hospital admission, median (range)	-	2 (0-45)
Diagnosis		
Influenza A	-	37 (86%)
Influenza B	-	1 (2%)
RSV	-	3 (7%)
SARS-CoV-2	-	2 (5%)
Location in hospital		
Ward	-	41 (95%)
ICU	-	2 (5%)
Level of Oxygen Support		
Oxygen Support	-	20 (47%)
None	-	23 (53%)

Individual summary

ID	Gender	Age range	Ethnicity	Number of visits	Days post hospital admission	Diagnosis	Severity	Peptide/HLA-A2-tetramer staining
1	F	55-59	Caucasian	2	1, 32	RSV	Ward	
2	M	70-74	Caucasian	2	0, 33	Influenza A	Ward	
3	M	35-39	Asian	1	2	Influenza A	Ward	
4	F	75-79	Caucasian	2	2, 45 (out of hospital)	Influenza A	Ward	Yes
5	F	25-29	Caucasian	1	1	Influenza A	Ward	
6	M	80-84	Caucasian	1	1	Influenza A	Ward	
7	M	85-89	Caucasian	1	2	Influenza A	Ward	
8	F	65-69	Caucasian	1	4	Influenza A	Ward	
9	M	65-69	Caucasian	1	2	Influenza A	Ward	
10	F	45-49	Caucasian	1	2	Influenza A	Ward	
11	M	50-54	Caucasian	1	2	RSV	Ward	
12	F	75-79	Caucasian	1	1	SARS-CoV-2	Ward	
13	F	40-44	Asian	1	2	RSV	Ward	
14	F	85-89	Caucasian	1	1	Influenza A	Ward	
15	M	60-64	Asian	2	1, 8	Influenza A	Ward	
16	M	70-74	First Nations	1	2	Influenza A	Ward	Yes
17	M	70-74	Caucasian	1	1	Influenza A	Ward	Yes
18	M	50-54	Caucasian	1	1	SARS-CoV-2	Ward	
19	F	65-69	Caucasian	1	3	Influenza A	Ward	
20	M	unk	Caucasian	1	1	Influenza A	Ward	
21	M	85-89	Caucasian	1	1	Influenza A	Ward	
22	M	70-74	Asian	1	5	Influenza B	Ward	
23	F	85-89	Caucasian	2	1, 3	Influenza A	Ward	
24	F	65-69	Caucasian	1	12	Influenza A	Ward, oxygen support	
25	M	70-74	Caucasian	1	3	Influenza A	Ward, oxygen support	
26	F	55-59	Caucasian	1	3	Influenza A	Ward, oxygen support	Yes
27	M	75-79	Caucasian	1	2	Influenza A	Ward, oxygen support	
28	M	80-84	Caucasian	1	1	Influenza A	Ward, oxygen support	Yes
29	M	45-49	Caucasian	1	2	Influenza A	Ward, oxygen support	Yes

30	F	30-34	Caucasian	1	4	Influenza A	Ward, oxygen support	
31	F	45-49	Caucasian	1	4	Influenza A	Ward, oxygen support	
32	M	65-69	Caucasian	1	2	Influenza A	Ward, oxygen support	
33	M	70-74	Caucasian	1	3	Influenza A	Ward, oxygen support	
34	F	55-59	Caucasian	1	1	Influenza A	Ward, oxygen support	
35	F	85-89	Caucasian	1	2	Influenza A	Ward, oxygen support	Yes
36	F	70-74	Asian	1	1	Influenza A	Ward, oxygen support	
37	M	65-69	Caucasian	1	1	Influenza A	Ward, oxygen support	
38	F	70-74	Caucasian	1	1	Influenza A	Ward, oxygen support	
39	M	50-54	Caucasian	1	2	Influenza A	Ward, oxygen support	
40	M	60-64	Caucasian	1	3	Influenza A	Ward, oxygen support	Yes
41	F	60-64	Asian	1	2	Influenza A	Ward, oxygen support	
42	M	50-54	Caucasian	1	3	Influenza A	ICU, Oxygen support	Yes
43	F	40-44	Caucasian	4	1, 5, 8, 15	Influenza A	ICU, Oxygen support	Yes

Also clarity on how data is comparable from one set to another in terms having been taken from the same patients.

IL18R1 expression in each cohort was analysed from the same patients and disease groupings as was performed for the analysis of differential *OLAH* expression described in our previous publication (Jia et al, Cell, 2024: PMID 3913778). This included the statistical methods used, models to control for factors and adjustments for multiple comparisons.

We have added the above sentences to the Methods (page 21).

The investigators should reframe their manuscript's novelty to incorporate existing published work by other researchers who have investigated this cohort (COVID/MIS-C/Severe viral illness) in children and adults and demonstrated upregulation of the IL-18 signalling pathway. They should include discussion of and their findings and how it integrates into the context of these other works.

We thank the Reviewer for this comment. We have now included additional discussion as recommended by the Reviewer (page 17):

“The IL-18 signalling pathway has previously been implicated as a correlate of severe disease. Indeed, recent studies investigating multisystem inflammatory syndrome in children (MIS-C), a life-threatening post-infectious sequelae of COVID-19 disease in paediatric cohorts, have shown that IL-18 concentrations were significantly increased in the plasma of patients with MIS-C as well as Hemophagocytic lymphohistiocytosis (HLH) relative to healthy controls⁴². Likewise, using a high dimensional mass cytometry approach to phenotype immune cells in MIS-C, Zhang *et al* further demonstrate elevated expression of IL-18R on CD16⁺ NK cells, monocytes, as well as TCR Vβ21.3⁺ CD4⁺ and CD8⁺ T cells in MIS-C patients⁴³.

Here, we further expand on this association between disease severity and the IL-18 signalling pathway and show that high IL-18R α expression is a feature of multiple severe respiratory infections, including highly pathogenic avian influenza H7N9 and respiratory syncytial virus (RSV), in addition to severe COVID-19. Consistent with previous findings by other groups, we further elaborate that IL-18R α expression on CD8⁺ and CD4⁺ T cells strongly correlates with disease severity and show that these can be recapitulated in a mouse model of mild and severe influenza. Collectively, our findings further support the use of IL-18R α as a potential biomarker for severe respiratory infections.”

Reviewer #3:

The paper was well done, although some points can be improved for the readability of the paper.

We thank the Reviewer for their supportive comments.

Firstly: in the introduction the role of IL-18 was explained in the context of inflammation mediated by viral infection. Although it is not clearly reported the dynamics and the biology which explain the link between IL-18 and OLAH.

We thank the Reviewer for this comment. Following the Reviewer’s recommendation (as well as the suggestion from Reviewer 1), we have performed experiments to investigate IL-18R α expression in OLAH^{-/-} mice.

We have investigated the IL-18R α expression on CD8⁺ T cells in *olah*^{-/-} mice on day 3 and day 6 following influenza virus infection. We have performed 3 independent experiments for day 3 (n=17 for WT, n=13 for *olah*^{-/-} mice) and 2 independent experiments for day 6 (n=10 for WT, n=10 for *olah*^{-/-} mice). Overall, our data showed no significant changes in either the frequency or the number of IL-18R α ^{hi} total CD8⁺ T cells in the absence of OLAH. Similarly, the number of IL-18R α ^{hi} effector CD8⁺ T cells remained unchanged in *olah*^{-/-} mice following influenza virus infections. Our results thus indicate that OLAH expression per se does not directly impact IL-18R α expression on CD8⁺ T cells in lungs of influenza-infected mice. This implicates that the high expression levels of both OLAH and IL-18R α , strongly linked with severe respiratory viral diseases in our human data (Fig. 1 and Fig. 2), are mediated via an independent mechanism.

We have now included the data in the revised version of the manuscript (Results; page 10):

“To understand any potential link between IL-18R α expression and OLAH, we investigated IL-18R α surface expression on CD8⁺ T cells in *olah*^{-/-} mice on day 3 and day 6 following influenza virus infection. We have performed 3 independent experiments for day 3 (n=17 for WT, n=13 for *olah*^{-/-} mice) and 2 independent experiments for day 6 (n=10 for WT, n=10 for *olah*^{-/-} mice). Overall, our data

demonstrated no significant changes in either the frequency or the number of IL-18R α ^{hi} total CD8⁺ T cells in the absence of OLAH (Supplementary Fig. 8). Similarly, the number of IL-18R α ^{hi} effector CD8⁺ T cells remained unchanged in *olah*^{-/-} mice following influenza virus infections. Our results thus indicate that OLAH expression per se does not directly impact IL-18R α expression on CD8⁺ T cells in lungs of influenza-infected mice. This implicates that high expression levels of both *olah* and *IL-18R α* , strongly linked with severe respiratory viral diseases in our human data (Fig. 1 and Fig. 2), are mediated via an independent mechanism.”

Supplementary Fig 8. IL-18R α expression on CD8⁺ T cells remains unchanged in influenza-infected *olah*^{-/-} mice. IL-18R α staining in the lungs of WT and *olah*^{-/-} mice following IAV infection. Frequencies and absolute numbers of IL-18R α ^{hi}CD8⁺ T cells and IL-18R α ^{hi}CD8⁺ effector T cells (CD62L^{lo}CD44^{hi}) were assessed on 3 and 6 following influenza virus infection. Data plotted are from 3 independent experiments for day 3 (n=17 for WT, n=13 for *olah*^{-/-}) and 2 independent experiments for day 6 (n=10 for WT, n=10 for *olah*^{-/-}). Bars represent mean ± SEM, statistically significant differences between groups were determined using unpaired Student’s t-test.

Moreover, in the final paragraph of the introduction (page 4 line 113) it is too much long the explanation of the results. I suggest to the authors to modulate this paragraph and move part of these in the discussion and better explain the hypothesis based on recent literature data.

We thank the Reviewer. As suggested by the Reviewer, we modified our Introduction not to discuss at length our results. We greatly appreciate the comment.

Concerning materials and methods, it could be of interest to add a kind of “flow chart” with the selection of patients and the enrollment criteria which better explain the cohort description.

We thank the Reviewer for the comment.

To clarify our human cohorts, in the modified version of our manuscript, we have now included:

- (1) A summary of the previously published human cohorts in Figure 1a, as below:

a

Cohort	Country	n	Age range	Analysis
A/H7N9 (Ref 15)	China	Fatal (4) Recovered (4)	53-88	Microarray (blood)
SARS-CoV-2 (Ref 15)	USA	Healthy (22) Infected (95) MIS-C (30)	0-21	Bulk RNASeq (blood)
pARDS (Ref 15, 21)	USA	RSV-infected (12) Non-RSV infection (5) Uninfected (6)	0-2	scRNASeq (tracheal aspirates)
Healthy (Ref 5)	USA	Uninfected (22)		Bulk RNASeq (blood)

Figure 1a. Summary of previously published disease cohorts, group sample sizes, age ranges and data used for IL-18R α analyses.

In addition, we have amended the text to provide greater clarity on which cohort is discussed as well as more information on the demographics or where they have been published.

Results (page 4):

“Here, in the same A(H7N9) cohort, our analysis of differentially expressed genes identified the *IL18R1* gene, encoding for IL-18R α receptor (CD218a; IL-18R α), as one of the three key genes highly expressed in fatal H7N9 patients early after hospital admission (within 6 days of hospital admission), alongside *OLAH* and *FLT3* (Fig. 1b)

“To determine whether the observed association between *IL18R1* levels and acute A(H7N9) disease severity could also be recapitulated in other respiratory diseases, we assessed *IL18R1* gene expression in a second disease cohort of patients hospitalized with acute SARS-CoV-2 infection and MIS-C (multisystem inflammatory syndrome in children)(Fig. 1a).”

Results (page 5):

“In comparison to samples from a cohort of healthy individuals that were used as controls (Fig 1a), *IL18R1* expression was significantly elevated in patients hospitalized with COVID-19 with no to minimal respiratory dysfunction (P=0.019).”

“In our third unrelated disease cohort²¹, single-cell RNA sequencing (scRNAseq) data were obtained from tracheal aspirates collected from children hospitalized with acute respiratory failure requiring endotracheal intubation, stemming from lower respiratory tract infection (LRTI)(Fig 1a).”

Materials and Methods (page 21):

“**H7N9, COVID-19 and pARDS patient cohorts.** A/H7N9-infected patients were admitted to the Shanghai Public Health Clinical Center between March and August 2013 and their clinical details have been previously published^{2, 13}. For our study, whole blood microarray samples from 8 patients¹⁵ were analyzed here according to DEGs for 4 patients who recovered (a73, a134, a20 and a9) and 4 patients with fatal outcomes (a118, a33, a131 and a22) at both early (within 6 days of hospital admission) and late stages of disease (at discharge or 21-22 days post-disease symptom onset). Informed consent was obtained from participants, and the study was approved and conducted under supervision by the SHAPHC Ethics Committee.”

“Blood from hospitalized patients aged 0-21 years was obtained as part of the Overcoming COVID-19 Study under IRB at Boston Children’s Hospital (IRB-P00033157) and the demographics of this cohort have been previously published ¹⁵.”

Materials and Methods (page 21):

“Samples from 22 uninfected, healthy individuals recruited at St Jude’s Children’s Research Hospital (Memphis, TN, USA) as part of the FLU09 cohort ⁵ were also used as controls in this study.”

“For pARDS analysis, we leveraged existing data from tracheal aspirate samples obtained from RSV infected children aged 0-2 recruited at LeBonheur Children’s Hospital (Memphis, TN, USA) ²¹ with 5 experiencing no to mild (n=5) pediatric ARDS (pARDS), 7 with moderate to severe pARDS and 5 with non-RSV related infection but with moderate to severe pARDS, and 6 control patients who were children without acute lower respiratory tract infection or lung injury.”

(2) A detailed Supplementary Table outlining demographics of the Melbourne cohort of patients hospitalised with respiratory viral diseases used in Figure 2 and Figure 3, as below:

Supplementary Table 1. Demographics of the acute viral respiratory hospitalised patient cohort in Melbourne.

Cohort Summary	Healthy	Patients
Number of individuals, n	17	43
Age, median (range)	22 (19-52)	68 (26-89)
Female, n (%)	14 (82%)	20 (47%)
Days post hospital admission, median (range)	-	2 (0-45)
Diagnosis		
Influenza A	-	37 (86%)
Influenza B	-	1 (2%)
RSV	-	3 (7%)
SARS-CoV-2	-	2 (5%)
Location in hospital		
Ward	-	41 (95%)
ICU	-	2 (5%)
Level of Oxygen Support		
Oxygen Support	-	20 (47%)
None	-	23 (53%)

Individual summary

ID	Gender	Age range	Ethnicity	Number of visits	Days post hospital admission	Diagnosis	Severity	Peptide/HLA-A2-tetramer staining
1	F	55-59	Caucasian	2	1, 32	RSV	Ward	
2	M	70-74	Caucasian	2	0, 33	Influenza A	Ward	
3	M	35-39	Asian	1	2	Influenza A	Ward	
4	F	75-79	Caucasian	2	2, 45 (out of hospital)	Influenza A	Ward	Yes
5	F	25-29	Caucasian	1	1	Influenza A	Ward	
6	M	80-84	Caucasian	1	1	Influenza A	Ward	
7	M	85-89	Caucasian	1	2	Influenza A	Ward	
8	F	65-69	Caucasian	1	4	Influenza A	Ward	
9	M	65-69	Caucasian	1	2	Influenza A	Ward	
10	F	45-49	Caucasian	1	2	Influenza A	Ward	
11	M	50-54	Caucasian	1	2	RSV	Ward	

12	F	75-79	Caucasian	1	1	SARS-CoV-2	Ward	
13	F	40-44	Asian	1	2	RSV	Ward	
14	F	85-89	Caucasian	1	1	Influenza A	Ward	
15	M	60-64	Asian	2	1, 8	Influenza A	Ward	
16	M	70-74	First Nations	1	2	Influenza A	Ward	Yes
17	M	70-74	Caucasian	1	1	Influenza A	Ward	Yes
18	M	50-54	Caucasian	1	1	SARS-CoV-2	Ward	
19	F	65-69	Caucasian	1	3	Influenza A	Ward	
20	M	unk	Caucasian	1	1	Influenza A	Ward	
21	M	85-89	Caucasian	1	1	Influenza A	Ward	
22	M	70-74	Asian	1	5	Influenza B	Ward	
23	F	85-89	Caucasian	2	1, 3	Influenza A	Ward	
24	F	65-69	Caucasian	1	12	Influenza A	Ward, oxygen support	
25	M	70-74	Caucasian	1	3	Influenza A	Ward, oxygen support	
26	F	55-59	Caucasian	1	3	Influenza A	Ward, oxygen support	Yes
27	M	75-79	Caucasian	1	2	Influenza A	Ward, oxygen support	
28	M	80-84	Caucasian	1	1	Influenza A	Ward, oxygen support	Yes
29	M	45-49	Caucasian	1	2	Influenza A	Ward, oxygen support	Yes
30	F	30-34	Caucasian	1	4	Influenza A	Ward, oxygen support	
31	F	45-49	Caucasian	1	4	Influenza A	Ward, oxygen support	
32	M	65-69	Caucasian	1	2	Influenza A	Ward, oxygen support	
33	M	70-74	Caucasian	1	3	Influenza A	Ward, oxygen support	
34	F	55-59	Caucasian	1	1	Influenza A	Ward, oxygen support	
35	F	85-89	Caucasian	1	2	Influenza A	Ward, oxygen support	Yes
36	F	70-74	Asian	1	1	Influenza A	Ward, oxygen support	
37	M	65-69	Caucasian	1	1	Influenza A	Ward, oxygen support	
38	F	70-74	Caucasian	1	1	Influenza A	Ward, oxygen support	
39	M	50-54	Caucasian	1	2	Influenza A	Ward, oxygen support	
40	M	60-64	Caucasian	1	3	Influenza A	Ward, oxygen support	Yes
41	F	60-64	Asian	1	2	Influenza A	Ward, oxygen support	
42	M	50-54	Caucasian	1	3	Influenza A	ICU, Oxygen support	Yes
43	F	40-44	Caucasian	4	1, 5, 8, 15	Influenza A	ICU, Oxygen support	Yes

Moreover, in my opinion it could be of interest also to perform a statistical analysis which explain that other variables (such as the comorbidities of the patients) do not affect the results related to “inflammatory status of the patients” and relative IL18R alpha expression.

We thank the Reviewer for this comment. P values were obtained from a model that controlled for days since symptoms onset, sex, whether a patient was previously healthy, steroid administration prior to sampling, bacterial co-infection, age, race and ethnicity, and were adjusted for multiple comparisons.

As per the Reviewer's request, we have provided plots that further demonstrate that age, gender and BMI (Supplementary Fig. 12) do not correlate with IL18R1 expression. We have described the data in Results (page 16):

“Our analysis of *GZMA*, *GZMB*, *PRF1*, *EOMES*, *KLRG1*, *CXCR1* and *IL-18R1* expression levels stratified according to these age ranges showed no clear correlation between age and expression of any of these genes (Supplementary Fig. 12).”

In my opinion it could be also interesting to deeply explain the results regarding the expression of IL-18 in NKT and NK cells. Due to the extreme correlation between innate and adaptive immunity in these kinds of patients and especially in the correlation with OLAH, please better explain in the discussion these data.

We thank the Reviewer for this comment. To answer the query, a significant correlation between IL-18R α expression on NK T cells and disease severity was found but not between IL-18R α expression on NK cells and disease severity (Supplementary Fig. 7f).

Following Reviewer's comment, we included more discussion on our data defining IL-18R α expression on NK and NKT cells (page 19):

In our manuscript, we focused predominantly on the role of IL-18R α expression on CD8⁺ T cells. However, we also found that IL-18R α was also highly expressed on innate immune cells such as NK cells, and NKT cells. Given the importance of IL-18R/IL-18 signalling in potentiating NK cell activity^{52,53}, we examined IL-18R α expression on NK and NK T cells throughout the course of mild and severe influenza disease. Consistent with previous findings showing NK cells constitutively express IL-18R α ⁵⁴, we found that the majority of NK cells expressed IL-18R α at baseline and expression levels did not substantially change throughout the acute and memory phases of IAV infection. Furthermore, while there was significantly higher frequencies of IL-18R α ⁺ NK cells in mild disease on day 6 post-infection, these differences were transient and were reversed by day 10. Interestingly, IL-18R α expression on NK T cells was more dynamic throughout the infection period. Frequencies of IL-18R α ⁺ NK T cells nearly doubled on day 3 post-infection but plummeted by day 6, before returning to baseline levels on days 10 and 28 post-infection as mice recovered, which may reflect the self-termination of NK T cell activity following primary activation, a phenomenon previously described in microbial infections⁵⁵. A significant correlation between IL-18R α expression on NK T cells and disease severity was found but not between IL-18R α expression on NK cells and disease severity.

I also suggest performing other analysis regarding these cellular clusters. The data regarding CD38, PD-1 and HLA-DR. In my opinion the role of these markers in the infection are still debates. This paper opens new frontiers in this field.

We thank the Reviewer for this comment. Following the Reviewer's comment, we have assessed CD38, PD-1 and HLA-DR in our Melbourne cohort of hospitalised patients with respiratory viral diseases. We found that increase in IL-18R α expression in patients was supported by co-expression of activation markers CD38 and HLA-DR on CD8⁺ T cells and influenza-specific CD8⁺ T cells in influenza-infected patients (Supplementary Fig. 3b and c).

We included these data in Results (page 6 and 7) and Supplementary Fig. 3b and c (below).

"Increase in IL-18R α expression in patients was supported by co-expression of activation markers CD38 and HLA-DR on CD8⁺ T cells (Supplementary Fig. 3b)."

"Expression of IL-18R α together with additional activation markers such as CD38, HLA-DR or PD-1 also increased in influenza-specific CD8⁺ T cells (Supplementary Fig. 3c)".

b Gated on CD8⁺ T cells

c

Supplementary Figure 3. IL-18R α expression on human monocytes and co-expression of activation markers. (b) Co-expression of IL-18R α , CD38 and HLA-DR on CD8⁺ T cell population from Melbourne patient cohort as described in (a). (c) Co-expression of IL-18R α , CD38, HLA-DR and PD-1 on unenriched CD8⁺ T cells, tetramer-enriched influenza-specific A2/M1₅₈⁺CD8⁺ T cells and unenriched CMV-specific A2/pp65₄₉₅⁺CD8⁺ T cells in influenza A patients. (b, c) Stacked columns indicate mean+SD. Statistical significance was analysed using Tukey's multiple comparisons test.

Although for the readability of the paper I suggest adding a paragraph in the introduction which better explains why the authors choose to explore the exhaustion state of the cells.

We thank the Reviewer for this question. We would like to clarify that PD-1 represents an activation marker during influenza virus infection. It is important to note that it is still unclear whether T cells are truly exhausted during acute influenza virus infection, and as such we investigated these markers.

We have clarified this in Results (page 9):

“We used PD-1 as a marker of T cell activation in acute influenza virus infection²⁶. It is important to note that it is still unclear whether T cells are truly exhausted during acute influenza virus infection.”

Point-by-point response to Reviewer's comments

The authors have revised the manuscript in response to reviewers' comments.

However, I have further questions and comments.

1. Reviewer #1's comment #2

I appreciate the authors' effort to address this comment and understand the result. If this is the case, I would like to recommend deleting the OLAH-related parts from this manuscript, including Figures 1C, 1F, 3C, 3F, and related contents. You may instead discuss OLAH as an independent biomarker in the Discussion. The story of IL-18Ra expression on CD8 T cells is sufficient to be the main topic of this manuscript, even without the data on its correlation with OLAH. The current manuscript places too much emphasis on the correlation between IL-18Ra and OLAH, which could mislead readers to assume a causal relationship between OLAH and IL-18Ra expression.

We thank the Reviewer for their comments. We have now revised the manuscript and removed OLAH related data from the suggested figures

2. Reviewer #1's comment #8

I understand the authors' response. However, if this is the case, many readers will likely wonder: if IL-18Ra-hi CD8 T cells produce IFN-g very well, is IFN-g beneficial or detrimental to the host? Please add discussion on the role of IFN-g produced by IL-18Ra-hi CD8 T cells during viral infections in the Discussion section.

We thank the Reviewer for this comment. We agree that the role of IFN γ and whether it is beneficial or pathological in the immune response to viral infections is highly interesting in the context of this manuscript. For this reason, we have discussed additional studies which have investigated the role of IFN γ in influenza virus infection. Furthermore, we have added on to this section and discussed whether excessive IFN γ production by IL-18Ra^{hi} cells may contribute to immune mediated pathology in severe influenza (please see lines 593-605 of the revised manuscript, also shown below):

“Our RNASeq data clearly demonstrated that amongst the most highly upregulated genes in IL-18Ra^{hi} T-cells was IFN γ , a finding validated in our functionality assays. Schmit et al. demonstrated that CD8⁺ T-cell-derived IFN γ exacerbated lung injury in influenza virus infection by promoting the recruitment of CCR2⁺ monocytes to the site of infection and supporting their differentiation into a pro-inflammatory, pathologic phenotype (50). In vitro co-culture studies similarly demonstrated that the combination of IFN γ and TNF derived from human CD8⁺ T-cells can lead to bystander lung damage by promoting downregulation of the epithelial Na,K-ATPase pump on uninfected epithelial cells, thus promoting fluid accumulation into the lung (51). Likewise, in a murine model of influenza-associated pulmonary aspergillosis, excessive IFN γ production was shown to have resulted in defective Th17-driven immune responses and impaired macrophage function, and that IFN γ ablation promoted improved disease outcomes (52). Hence, CD8⁺ T-cells with high levels of IL-18Ra may contribute to influenza disease severity through pathologic secretion of IFN γ .”

Reviewer #2 (Remarks to the Author):

The manuscript is much improved and I appreciate the clarity in methods and discussion in the revised manuscript. I have no further comments

We would like to thank Reviewer 2 for their kind comments.

Reviewer #3 (Remarks to the Author):

The paper has been significantly improved, and the authors have followed the reviewers' suggestions. In my opinion the paper can be accepted for publication.

We thank Reviewer 3 for their comments and helpful contributions to the revised manuscript.